# *Aspergillus brasiliensis* E_15.1: A Novel Thermophilic Endophyte from a Volcanic Crater Unveiled through Comprehensive Genome-Wide, Phenotypic Analysis, and Plant Growth-Promoting Trails

**DOI:** 10.3390/jof10080517

**Published:** 2024-07-25

**Authors:** Amanda Martirena-Ramírez, José Germán Serrano-Gamboa, Yordanis Pérez-Llano, Claribel Orquídea Zenteno-Alegría, Mario León Iza-Arteaga, María del Rayo Sánchez-Carbente, Ana María Fernández-Ocaña, Ramón Alberto Batista-García, Jorge Luis Folch-Mallol

**Affiliations:** 1Centro de Investigación en Biotecnología, Universidad Autónoma del Estado de Morelos, Cuernavaca 62209, Morelos, Mexico; amanda.martirena@uaem.edu.mx (A.M.-R.); jose.serrano@uaem.mx (J.G.S.-G.); mario.iza@uaem.edu.mx (M.L.I.-A.); maria.sanchez@uaem.mx (M.d.R.S.-C.); 2Centro de Ciencias Genómicas, Universidad Autónoma del Estado de Morelos, Cuernavaca 62209, Morelos, Mexico; yordaniz.perezlla@uaem.edu.mx; 3Centro de Investigación en Dinámica Celular, Instituto de Investigación en Ciencias Básicas y Aplicadas, Universidad Autónoma del Estado de Morelos, Cuernavaca 62209, Morelos, Mexico; claribel.zenteno@uaem.edu.mx; 4Facultad de Ciencias Químicas e Ingeniería, Universidad Autónoma del Estado de Morelos, Cuernavaca 62209, Morelos, Mexico; 5Departamento de Biología Animal, Biología Vegetal y Ecología, Facultad de Ciencias Experimentales, Universidad de Jaén, 23071 Jaén, Spain; amocana@ujaen.es

**Keywords:** *Aspergillus brasiliensis*, endophytic fungus, extreme environments, thermophilic, phylogenomic analysis, plant growth-promoting fungus

## Abstract

Thermophilic fungi have been seldom studied despite the fact that they can contribute to understanding ecological mechanisms of adaptation in diverse environments and have attractive toolboxes with a wide range of biotechnological applications. This work describes for the first time an endophytic and thermophilic strain of *Aspergillus brasiliensis* that was isolated in the crater of the active volcano “El Chichonal” in Mexico. This strain was capable of surviving in soil with a temperature of 60 °C and a pH of neutral acidity, which preluded a high thermostability and a potential in industrial application. The complete genome of *A. brasiliensis* E_15.1 was sequenced and assembled in 37 Mb of genomic DNA. We performed a comprehensive phylogenomic analysis for the precise taxonomic identification of this species as a novel strain of *Aspergillus brasiliensis*. Likewise, the predicted coding sequences were classified according to various functions including Carbohydrate-Active Enzymes (CAZymes), biosynthetic gene clusters of secondary metabolites (BGCs), and metabolic pathways associated with plant growth promotion. *A. brasiliensis* E_15.1 was found to degrade chitin, chitooligosaccharides, xylan, and cellulose. The genes to biosynthesize clavaric acid (a triterpene with antitumor activity) were found, thus probably having antitumor activity. In addition to the genomic analysis, a set of enzymatic assays confirmed the thermostability of extracellular xylanases and cellulases of *A. brasiliensis* E_15.1. The enzymatic repertoire of *A. brasiliensis* E_15.1 suggests that *A. brasiliensis* E_15.1 has a high potential for industrial application due to its thermostability and can promote plant growth at high temperatures. Finally, this strain constitutes an interesting source of terpenoids with pharmacological activity.

## 1. Introduction

Extreme environmental conditions can shape specific microbial communities. Temperature constitutes a significant factor influencing biomolecule functions and the preservation of their biological structures. The upper-temperature threshold for eukaryotes has been documented at 62 °C, [1] and only a limited number of fungal species are recognized as genuinely thermophilic [2] Thermotolerant and thermophilic fungi (TTF) grow above 45–50 °C; the latter are unable to grow below 20 °C. The growth temperature record (up to 60 °C) belongs to *Chaetomium thermophilum*, isolated from dung/compost [2]. More than 30 species have been identified as TTF [3,4].

Endophytic fungi are organisms that inhabit plant tissues which they colonize through balanced antagonistic interactions, that is, they need to activate virulence mechanisms that trigger a defensive response in plants [5]. Plant Growth-Promoting Fungi (PGPF) play a central ecological role by contributing to both plant health and soil fertility. Among the mechanisms used by PGPF are enhancing nutrient availability (such as nitrogen, phosphorus, potassium, magnesium, iron, etc.), production of plant hormones, and augmenting water absorption rates. In addition, some PGPF also induce systemic resistance (ISR) against phytopathogens. The well-known fungal genera *Aspergillus*, *Fusarium*, *Penicillium*, *Piriformospora*, *Phoma*, and *Trichoderma* are the most frequently reported PGPF [6].

Using Plant Growth-Promoting Fungi in sustainable agriculture represents an environmentally friendly alternative since they can contribute by reducing the use of polluting compounds such as chemical fertilizers and pesticides. PGPF can also increase resistance and tolerance in plants against biotic and abiotic stresses. Thermophilic and thermotolerant PGPF are frequently associated with plants that live in environments with high temperatures such as geothermal soils. Likewise, they have been successfully applied as agents both for biofertilization and to confer tolerance to heat stress in different plant species. An example of the above is the fungal endophyte *Curvularia protuberate*, which was isolated from *Dichanthelium lanuginosum*, a grass that thrives in Lassen Volcanic and Yellowstone National Parks [7]. This fungus has the capacity to confer resistance to elevated soil temperatures within the range of 38–65 °C both in the host plant and many other crop plants such as tomato, watermelon, and wheat [8].

Aspergilli are cosmopolitan ascomycetes and one of the most studied fungal genus after yeasts. Aspergilli have been frequently studied regarding degradation of plant biomass, secondary metabolism, and signal transduction [9]. Due to its versatility and relative ease of cultivation, Aspergilli have been used for several decades in various biotechnological applications that include foods, pharmaceuticals, materials, agroindustry, and environmental issues. Recently, this genus has gained popularity as a cell factory to produce recombinant proteins due to its native ability to secrete proteins with high yields [10]. Some *Aspergillus* species have been recognized as thermophilic or have been documented to be capable of producing thermostable enzymes [11,12,13,14]. Likewise, several plant growth-promoting traits have been observed in Aspergilli, including the production of extracellular phytases, which mineralize phosphate from inaccessible organic sources, siderophores, and phytohormones such as IAA [15,16,17].

Whole-genome studies constitute a suitable approach for the identification of sequence-based features associated with capabilities such as plant growth-promoting traits. In fungi, there is a growing tendency to complement research on plant growth promotion with genomic studies [18,19].

However, functional genomics in fungi still presents certain limitations such as the difficulty in reconstructing sequences at a chromosomal level or the identification of novel proteins, many of which still lack a functional assignment. In addition to the above, the taxonomic classification of new fungal species requires more solid evidence, supported by genomic coherence parameters. In the past, assignments based on DNA barcoding of popular markers such as the Internal transcribed spacer (ITS) or translation elongation factors (TEF) were considered to be robust and accurate; however, given the growing number of documented species, the accuracy and resolution of such markers are unable to avoid ambiguity when identifying new taxa [20].

Fungi from extreme environments, especially those isolated from high-temperature exposed regions (i.e., volcanos), offer promising avenues for developing innovative strategies to promote plant growth in the context of global climate change [7]. These fungi have adapted to thrive under harsh conditions, such as high temperatures, which makes them inherently resilient and capable of withstanding environmental stresses that are increasingly prevalent due to climate change [21]. This resilience can be harnessed to enhance plant tolerance to heat stress, which is critical as rising global temperatures threaten agricultural productivity. Moreover, thermophilic and thermotolerant fungi often possess unique metabolic pathways and enzymes that facilitate nutrient uptake and stress resistance in plants [22]. For instance, these fungi can produce siderophores and heat-stable enzymes, such as phytases, that improve nutrient availability and uptake even under adverse conditions. Additionally, the production of phytohormone-like compounds by these fungi can promote plant growth and development, further enhancing their potential as biofertilizers [7]. By leveraging the robust capabilities of these extremophiles, it is possible to develop sustainable agricultural practices that not only improve crop yields but also reduce reliance on chemical fertilizers and pesticides, mitigating their environmental impact. Thus, the study and application of fungi from extreme environments represent a forward-thinking approach to addressing the challenges posed by global climate change in agriculture.

In this work, we present a thorough phylogenomic analysis to determine the identity of an endophytic fungus of the genus *Aspergillus.* This fungus was able to solubilize phosphate as well as produce siderophores and phytohormones. These traits were complemented with the functional annotation of the coding sequences of its whole genome. Plant growth-promoting traits, CAZymes, and Biosynthetic gene clusters (BGCs) were especially interesting. To date, all the reported *Aspergillus brasiliensis* strains are rhizospheric and mesophilic. This is the first report of an endophytic and thermophilic *A. brasiliensis* strain, which, besides being a good plant growth promoter, protects the plant from heat stress. It produces thermostable enzymes, which could be of potential biotechnological use. Secondary metabolite genes, such as the antitumoral compound clavaric acid were also found. The experimental data agree with the phylogenomic study of this novel strain.

## 2. Materials and Methods

### 2.1. Isolation of Fungal Strains

The Chichonal volcano (17.36 °N, 93.23 °W) is located in the northwestern state of Chiapas, Mexico, at an altitude of 1100 m above sea level and it is still active. A lake has formed inside the crater with huge temperature and pH variations from the volcano vent to other more distant places within the crater. Three endophytic strains were isolated, one from an *Andropogon* sp. (as determined by its morphological characters) and two from an (until now) unidentified dicotyledonous plant (probably from the genus *Gaulheria* sp., according to the PlantNet application; manuscript in process). From the three isolates, strain E_15.1 was selected for further characterization due to its preliminary characterization (see below).

Fresh roots were collected from the plants at one specific location (designated as SP15) within the Chichonal volcano crater (Figure 1).

It is worth noting that the *Andropogon* sp. plants were alive and submerged in the shore of the lake inside of the crater. Three samples of fresh roots were taken for each set of specimens. For the isolation of endophytic fungi, a surface sterilization procedure was carried out to avoid the growth of rhizospheric fungi [23]. To disinfect the collected root samples, they were first surface cleaned with running tap water for ten minutes, followed by a rinse with sterile distilled water for five minutes. Subsequently, the roots were immersed in a 70% *v/v* ethanol solution for 3 minutes in a laminar flow cabin, and then treated with a 4% *v/v* NaClO solution supplemented with Tween 80^®^ (0.1% *v/v*) (Thermo Fisher Scientific, Cleveland, OH, USA) for 5 minutes. Finally, the roots were rinsed with sterile distilled water for 1 min and dried on sterile paper towels. Pieces of 4 × 0.5 cm and 1.5 × 0.5 cm were cut from the disinfected tissues under aseptic conditions and placed on potato dextrose agar (PDA; Sigma, St. Louis, MO, USA) culture medium. The Petri dishes were supplemented with 50 μg/mL of chloramphenicol (Clor) and 100 μg/mL of ampicillin (Am) to prevent bacterial growth. The “leaf imprints” test was used as a control to ensure the success of the surface sterilization process [23]. For this, the disinfected roots were gently pressed against a Petri dish with PDA Am/Clor medium, and after 1 min, they were removed. The surface sterilization process was considered effective if no growth was observed on these dishes. To emphasize the endophytic nature of the fungi, only the tips of the hyphae protruding from the roots were selected. The tips of individual hyphae were subcultured and transferred three times to Petri dishes with PDA Am/Clor medium to ensure the purity of the isolates.

### 2.2. Determination of Optimal Growth Temperature and Morphological Characterization

The optimal growth temperature was determined by growing the isolates in a liquid MEA medium (Merck KGaA, Darmstadt, Germany). Media were adjusted to 0.2 (600 nm) and were incubated at 28, 40, and 70 °C, with shaking. Samples were taken at 24, 36, 48, 60, 72, 84, and 96 h and the optical density was read at 600 nm, in triplicate, at the 3 temperatures in a Nanodrop 2000 spectrophotometer (Thermo Scientific, Cleveland, OH, USA). The speed or growth rate (µ) was calculated with the following formula:µ = Xf/Xi × (t2 − t1)
where

µ = speed or growth rate (h−1);

Xf = final biomass concentration (given in optical density);

Xi = initial biomass concentration;

t2 = final time in hours;

t1 = initial time in hours.

The growth of the endophytic fungus E_15.1 was also evaluated by plate growth, on a solid medium (MEA), inoculated with a 5mm portion of fresh mycelium, with incubation at 28 and 40 °C. The diameter of the colony was recorded and monitored for 96 h. The results were expressed as mycelial growth (cm).

To conduct the morphological characterization of strain E_15.1, 9 µL of a precultured liquid medium was inoculated, forming a triangular pattern on three solid media types: PDA, MEA, and CYA (Appendix A). The cultures were observed for the growth of the respective fungus on the three different media, and photographs were taken at intervals of 1, 2, 3, 7, and 14 days. For the microscopic analysis, a culture loop was employed, and a sample was placed on a microscope slide. A drop of lactophenol cotton blue stain (a solution containing 20% lactophenol and 10% cotton blue) was added for examination under a phase-contrast and bright-field optical microscope. This allowed the description of the type of mycelium and the structures involved in asexual reproduction.

### 2.3. Culturing and Extraction of Genomic DNA

E_15.1 strain was cultured in solid MEA medium for 5 days at 28 °C. The mycelium was collected and genomic DNA extraction was performed following the protocol proposed by Kuhad et al. (2004) [24]. To assess the quality of the DNA, agarose gel electrophoresis was conducted, and the sample was measured using a “NanoDrop” spectrophotometer (Thermo Scientific), before it was sent to Macrogen Inc., Seoul, Korea, for whole genome sequencing.

### 2.4. Genome Sequencing, Assembly, and Functional Annotation

For the whole genome sequencing of strain E_15., the Illumina MiSeq platform was adopted (Illumina Inc.San Diego, CA, USA). The raw sequence data coming from the high throughput sequencing pipelines were applied to the program FastQC (http://www.bioinformatics.babraham.ac.uk/projects/fastqc, accessed on 1 June 2024) and MultiQC (v 1.14) [25], which are tools for quality control of the sequencing. Two versions of genomic assemblies MEGAHIT (v1.2.9) [26] and SPAdes were created [27], and then assembly fusion was performed with Quickmerge [28], generating a more contiguous final assembly. Finally, the contigs were polished using Pilon (v1.22) [29] and with the RagTag tool (v2.1.0) [30]. For an assembly at the level of the scaffold, the SeqKit tool (v2.4.0)[31] was employed to generate the assembly statistics including N50 and N90. After having the complete genome data of strain E_15.1. the predicted set of coding DNA sequences (CDS) performed with the AUGUSTUS tool (v3.2) [32]. Protein sequences were annotated using the BLASTKOALA tool (v3.0) [33]. This was performed to identify functional orthologs associated with different metabolic pathways based on the Kyoto Encyclopedia of Genes and Genomes categories (KEGG). Similarly, the predicted set of genome proteins was annotated using the PANNZER2 web server (URL: http://ekhidna2.biocenter.helsinki.fi/sanspanz/, accessed on 27 April 2023) [34] to classify the protein set into functional categories according to the Gene Ontology (GO). Finally, the assignation of Cluster of Orthologs Groups (COG) was performed by the eggNOG-mapper in its web server (URL: http://eggnog-mapper.embl.de/, accessed on 2 October 2023) [35].

For comparative genomic analysis, the genomes of ten Aspergilli species were downloaded from the NCBI database, including (i) strains beneficial for plants—*A. brasiliensis* (CBS101740); (ii) strains used as biological control for plants—*A. piperis* (CBS112811); (iii) pathogenic strains—*A. fumigatus* (AF 293); *A. viridinutans* (IFM47045) and *A. pseudoviridinutans* (IFM55266); and (iiii) enzyme-producing strains with important industrial applications—*A. tubingensis* (WU223-L), *A. vadensis* (CBS113365), *A. luchuensis* (IFO 4308), *A. aculeatus* (ATCC 16872), and *A. carbonarius* (ITEM 5010).

Furthermore, fungal proteins were annotated according to the Carbohydrate-Active enZymes (CAZy) database version 12 (release date: 2 August 2023) downloaded from the dbCAN web server (URL: https://bcb.unl.edu/dbCAN2/, accessed on 1 June 2024) [36]. In order to identify sequences related to plant growth promotion, proteins from the E_15.1 genome were annotated using HMMER3 (v 3.3.2) [37] using a set of Hidden Markov Model (HMM) profiles corresponding to protein families in the Pfam database (release 36 on 12 September 2023) [38] associated with relevant functions (see Table 3) employed for this purpose. Finally, a global annotation of the whole set of predicted proteins in the E_15.1 genome was carried out by aligning them against the NCBI non-redundant protein database (nr) using DIAMOND (v. 0.9.14.115) [39].

### 2.5. Phylogenetic Analysis

Phylogenetic analyses were conducted using 68 genome assemblies of the *Aspergillus* genus, with 2 genome assemblies of the *Penicillium* genus as an outgroup. Various phylogenetic analysis programs were executed, starting with JolyTree (v.1.1b.191021ac) [40], which employs a distance-based alignment-free procedure to infer phylogenetic trees. Subsequently, the OrthoFinder program (v2.5.4) [41] was used for high-precision phylogenetic orthology inference and provides phylogenetic inferences of orthologs. Input data for this program included predicted protein sequences generated by the Augustus tool (v3.2) [32]. Finally, the UFCG program (v1.0.3) (Universal Fungal Core Genes), which utilizes a database of canonical genes and core fungal genes [42], was employed for genome analysis and inferring the species phylogenomic tree.

We conducted species delimitation using distance-based models (Bayesian Poisson tree processes (bPTP) and Multirate Poisson tree processes (mPTP)). Furthermore, tests were carried out for test species delimitation using bPTP version 0.51 [43], implemented in Python (v. 3.7.6), and adjusted to accept non-ultrametric trees as input files (https://github.com/zhangjiajie/PTP, accessed on 1 June 2024). Similarly, the mPTP model (https://github.com/Pas-Kapli/mptp, accessed on 1 June 2024) models speciation events considering lineage-specific coalescence rates and a speciation parameter.

The phylogenetic trees (JolyTree, OrthoFinder, and UFCG) generated in Newick format were used as inputs in separate runs. Convergence was assessed through 106 iterations of Markov Chain Monte Carlo (MCMC) chains, with the number of seeds set to 4.

Subsequently, we evaluated the mutational genomic distance (D) using the Mash program v2.3 [44] and the average nucleotide identity (ANI) was calculated using FastANI v1.33 [45].

### 2.6. Identification of Metabolic Pathways and Specialized Metabolites’ Synthesis Genes

Genomic analyses were performed for the determination of metabolic pathways associated with PGP traits (gibberellin, indole acetic acid (IAA) production, siderophore production, and iron and phosphate uptake) with KEGG and complemented with KofamKOALA (profiles released 3 October 2023) [46]. Additionally, the full genome of the compared eleven strains was submitted to fungal version AntiSMASH v6.0.1 [47] for secondary metabolite biosynthetic gene cluster (BGC). Furthermore, it compared the genetic diversity of the biosynthetic clusters of clavaric acid, a triterpene with antitumor activity due to its ability to inhibit Farnesyl-protein transferase-producing BGC components against the antiSMASH database.

### 2.7. In Vitro Assays for PGP Traits and Hydrolytic Enzymatic Activity

Different culture media were used to determine seven PGP traits and some hydrolytic activities of strain E_15.1. The activity of cellulases, xylanases, and chitinases was evaluated in solid Vogel’s minimal medium supplemented with 1% carboxymethylcellulose (CMC), beechwood xylan, or colloidal chitin as the sole carbon source for each assay. Clear halos around the colonies indicated the presence of cellulases, xylanase, or chitinase activity [48]. The siderophore production detection test involved inoculating double-layered agar plates with Chromium Azurol S (CAS), following Louden et al. (2011) [49]. The appearance of a yellow-orange or purple halo around the colony was considered positive siderophore producers. The phosphate solubilization assays were conducted using the PKV (Pikovskayas agar) culture media, where growth is associated with the capacity to use inorganic phosphate in the form of Ca_3_(PO_4_)_2_ as a sole phosphate source. This method is considered to show positive results when a transparent halo is observed in the plates. The production of Indole 3 acetic acid (IAA), was determined following the methodology described by Ignatova et al. (2015) [50] with some modifications. The isolates were cultured in MEA liquid medium at pH 5.6 with 400 or 1000 μg/mL of L-tryptophan as the precursor for IAA biosynthesis. Cultures without tryptophan were used as negative controls. The cultures were incubated at 28 °C for three days. The supernatant (0.5 mL) was taken and mixed with 0.5 mL of Salkowski’s reagent (2 mL of FeCl_3_ (0.5 M) and 98 mL of H_2_SO_4_ (38%). They were then incubated for 30 min at room temperature in the dark. The color intensity of the reaction was measured at a wavelength of 530 nm, using an Epoch microplate spectrophotometer (BioTek, Sta. Clara, CA, USA). Equivalent mixtures with non-inoculated media served as blanks for the spectrophotometry readings. A calibration curve was performed with standard IAA (0–40 µg/mL) (Sigma Catalog No. 6505-45-9) to quantify the concentration of IAA. Finally, the evaluation of gibberellin production was determined using a protocol described by Candau et al. (1992) [51]. Czapek culture medium (without inoculation) was used as a negative control and tubes to which 2 µg gibberellic acid (GA3) (Sigma, St.Louis MO, USA) were added as a positive control. Samples showing green fluorescence were considered positive for gibberellin production.

### 2.8. Plant Responses to E_15 Interaction in Greenhouse Conditions

The effect of strain E_15.1 on the growth of tomato plants was assessed. The plants were cultivated in a greenhouse from the early vegetative stage to the onset of the reproductive phase (see below) between March and May 2023, a period outside the optimal tomato cultivation season (July–September). For this study, 20 plants were utilized for each treatment. Two additional groups were included for comparison: one group was inoculated with sterile distilled water (Negative Control), and the other group was inoculated with *Trichoderma atroviride* (Positive Control). All the analyzed plants originated from Saladette variety tomato seeds, specifically the Hortaflor brand, and were initially germinated in trays with 200 compartments. Two-week-old plants were then transplanted into pots containing autoclave-sterilized Peat moss^®^ substrate. Seven days after transplantation into the pots, the plants were inoculated with a suspension of 1 × 10^6^ spores/mL of strains E_15.1 and *Trichoderma atroviridae*, each suspended in 4 mL of sterile distilled water. Control plants were treated with 4 mL of sterile distilled water. The inoculation took place at the substrate level, after it had been previously watered and was still moist. Subsequently, the plants were placed in a greenhouse of the Faculty of Agricultural Sciences (UAEM, Morelos, México), where they were exposed to natural light from March (12 light/12 dark, approximately) to May (14 light/10 dark, approximately). Regular watering was carried out every third day. This study assessed the effects of growth promotion mediated by strain E_15.1 on the following parameters: stem length (cm),—from the longest apical leaf to the root neck; root length (cm)—measurement from the apex of the primary root to the neck root of each sampled plant; and biomass/dry weight (g)—plants were dried in an oven at 60 °C for 72 h until constant dry weight was achieved and fresh weight biomass (g) was immediately measured after collecting the roots.

### 2.9. Effect of Temperature on the Activity of Xylanases and Cellulases by E_15.1 Strain

The extracellular enzymatic activity of the E_15.1 strain was measured based on the amount of released reducing sugars using the 3,5-dinitrosalicylic acid (DNS) method [52]. Calibration curves were performed using glucose and xylose, as standards. One unit of xylanase and cellulase activity (U) is defined as the amount of enzyme releasing 1 µmol reducing sugar per min. For the assessment of cellulase activity, a 1% solution of carboxymethylcellulose (CMC) was used as a substrate, while a 1% solution of beechwood xylan (Sigma) was used for xylanase activity determination. These substrates were dissolved in a 50 mM citrate buffer solution at pH 5 (cellulase activity) and 30 μL of sodium phosphate buffer (0.1 M) at pH 8.0 (xylanase activity). The enzymatic reaction was established, comprising 250 µL of the substrate solution, 200 µL of a 50 mM citrate buffer solution at pH 5 or 30 μL of potassium phosphate buffer (0.1 M) at pH 8.0, and 50 µL of the culture supernatant of strain E_15.1, grown in Vogel’s minimal medium supplemented with 2% wheat straw as the sole carbon source. To determine the temperature optima, the reaction mixtures were incubated at various temperatures (30, 40, 50, 60, and 70 °C) for 15 min. Samples of 25 µL were withdrawn, mixed with 25 µL of DNS, boiled for 5 min, and allowed to cool for 5 min to terminate the reaction. Subsequently, 50 µL of distilled water was added. The absorbance was measured using a spectrophotometer at 540 nm. The reaction blank consists of a water-adjusted volume of the reaction mixture without the culture supernatant. Each sample was analyzed in three experimental replicates. The thermostability of hydrolytic enzymes was also tested by incubating the supernatants of each condition at 70, 80, and 90 °C for one hour with the same reaction proportions described above. The residual enzyme activity (RA) was calculated from the relationship between the enzyme activity after treatment and the initial activity without treatment. Its calculation was carried out using the following equation reported by Riener et al. (2009) [53] in percentage terms:RA (%) = (At/A0) × 100

In this equation At is the activity of the enzyme after heat treatment and A0 is the activity of the enzyme without heat treatment.

### 2.10. Statistical Analysis

For the greenhouse plant growth parameter experiment of *S. lycopersicum*, the treatments consisted of plants inoculated with the endophytic fungus (E_15.1) strain, a positive control (plants inoculated with *Trichoderma atroviridae*), and a negative control (plants inoculated with sterile distilled water). The dependent variables were root length (RL), stem height (SH), total plant fresh weight (FW), and total dry weight (DW). Experimental units were randomly distributed. Results were analyzed using a one-way ANOVA, and assumptions on residuals were checked. All analyses were performed using the R 4.3.2. generic version (R Core Team 2017) and Python (3.12.0) version. A Tukey test was used for mean separation. The analyses were conducted at a significance level (* *p* ≤ 0.05, ** *p* ˂ 0.01).

To determine the similarities between the genomes of the compared strains regarding genomic functional annotations, a non-metric multidimensional scaling (NMDS) analysis was conducted. All analyses were performed using the VEGAN package (vegan_2.6-6.1.tar.gz) in the R programming language version 3.2.2 (R Core Team 2017). The data were normalized, and a Euclidean distance was chosen to carry out the analysis.

## 3. Results

### 3.1. Isolation, Optimal Growth Temperature, and Morphological Characterization

Endophytic fungal isolates obtained from disinfected root tissues of an *Andropogon* sp. were cultivated as described in the methodology, and mycelial growth was observed after seven days of incubation (Figure 2A). No mycelial growth was observed in the control root imprint dishes (Figure 2B).

The endophytic fungus E_15.1 grew in solid medium at the lowest temperature tested (28 °C) and at the highest (40 °C) (Figure 3A,B), showing a higher growth rate at 40 °C, where their mycelial growth at 96h ranged between 7.5 and 8 cm (Figure 3B), while for this same time, mycelial growth at 28 °C was 5 cm (Figure 3B). Similarly, it grew in liquid medium at the three temperatures evaluated, showing a higher growth rate at 40 °C (0.06 h^−1^), which classifies it as thermophilic (Figure 3C). Morphological characterization was performed, considering the morphology exhibited by the mycelium (size, shape, colony color) in the different culture media (CYA, MEA, and PDA) used for morphological characterization at seven days of incubation. The endophytic fungus E_15.1 was characterized by the presence of large colonies on Day 7 in all three types of culture media: MEA, PDA, and CYA (Figure 2D–F). The colonies were round with filamentous edges in all three media (Figure 2D–F). In PDA and CYA, the colony surface appeared powdery (Figure 2E,F), while in MEA, it appeared cottony (Figure 2D). MEA, PDA, and CYA exhibited black pigmentation with white tones on the front and beige tones on the back (Figure 2D–F). PDA displayed beige pigmentation with brown and white tones, and it was yellow on the back (Figure 2F). In CYA, a surface with regular radiating lines and a wrinkled texture with cream coloration on the back was observed (Figure 2E). In all three media, the colonies exhibited a convex elevation (Figure 2D–F). The microscopic characterization with lactophenol staining of mycelium grown in MEA culture medium after 5 days of incubation at 28 °C, the presence of conidiophores with phialides and conidia with light green pigmentation could be observed (Figure 2G,I,K) as well as septate hyphae. (Figure 2H,J,L).

### 3.2. Genome Sequencing, Assembly, and Functional Annotation

The genome of strain E_15.1 was sequenced using the Illumina MiSeq platform, yielding a total genome size of approximately 36.9 Mb with an N50 length of 1,968,182 bp (Appendix A). This genome size is comparable to that of *A. brasiliensis*, *A. carbonarius*, *A. aculeatus*, and *A. pseudoviridinutans*, indicating a significant difference from other *Aspergillus* species (Table 1). A total of 17,182,281 paired-end reads were obtained, each with a sequence length of 150 bp. Among these, 10,926 protein sequences (99.4%) aligned successfully with the nr database, with 8553 (77.81%) showing the highest similarity to proteins from *Aspergillus brasiliensis* CBS 101740. This high level of similarity strongly suggests that strain E_15.1 is closely related to *A. brasiliensis* in taxonomic terms.

A total of 9101 proteins (82.79%) were assigned to at least one of the three main GO categories (Figure 4A). We also identified 10,361 sequences (94.25%) associated with COG functions (Figure 4B). Interestingly, in both COG and GO annotation, abundant protein sequences associated with secondary metabolism were observed (Figure 4A,B).

Similarly, a total of 4130 proteins (37.57%) were designated as KEGG orthologs (KO). The functional annotation by KEGG of the proteins predicted by AUGUSTUS from the fungal genome E_15.1 and 10 strains belonging to the *Aspergillus* genus in relation to key pathways of secondary metabolism, xenobiotic degradation, lipid, and carbohydrate metabolism, as well as orthologs associated with amino sugar and nucleotide sugar metabolism, is depicted in Figure 5.

The most abundant KEGG categories were represented by glycan biosynthesis and metabolism, carbohydrate and lipid metabolism, and metabolism of terpenoids and polyketides (Figure 5A). Regarding the orthologs related to the amino sugars metabolism, the largest number of orthologs was represented by K01183 (chitinase), followed by K00698 (chitin synthase) (Figure 5B)—results that could be related to the degradation mainly of chito-oligosaccharides perhaps involved in cell wall biosynthesis in E_15.1 strain.

A total of 919 proteins (8.36%) were annotated as CAZymes or proteins related to such functions, such as carbohydrate-binding modules (CBM) and auxiliary activities (AA) (Figure 6A). Additionally, they were categorized into CAZy protein families with catalytic activities on various substrates (Figure 6B). Approximately one-third of the predicted CAZy proteins were linked to the degradation or chemical modification of specific substrates, with the highest proportion observed in relation to chito-oligosaccharides (Figure 6B).

### 3.3. Phylogenetic Analysis

Phylogenomic analysis based on mutational distances resulting from the comparison of 68 fungal genomes from RefSeq categories within the *Aspergillus* genus in JolyTree is shown in Figure 7. The phylogenomic relationship of strain E_15.1 was found to be closely related to the genome of strain GCA001889945.1 (CBS 101740T), classified as *A. brasiliensis*. The genetic orthology analysis (Figure 8) and the UFCG (Universal Fungal Core Genes) analysis (Figure 9) similarly showed that the E_15.1 strain is closely related to the *A. brasiliensis* strain. OrthoFinder results showed that there are a total of 15,169 orthologous protein groups present among the predicted proteomes and a total of 1119 orthologous proteins existing as a single copy.

All phylogenetic trees yielded strong evidence for a monophyletic clade consisting of *A. brasiliensis*, allowing us to classify E_15.1 as *A. brasiliensis*.

Furthermore, species delimitation tests in both algorithms (bPTP and mPTP) for all phylogenetic trees aim to determine the transition point from one speciation process to another coalescence process. In our case, the E_15.1 strain is constrained to a single coalescence rate across all trees with *A. brasiliensis* by accommodating specific intraspecies coalescence rates. Furthermore, a *k-mer* analysis using FOCUS software (v 1.8) showed that E-15.1 and *A. brasiliensis* IFM_66951 share 90.7% identical *k-mers*, indicating that E-15.1 is an *A. brasiliensis* strain. This software “breaks” the genomes into small fragments (6–9 nucleotides), allowing the detection of alleles and mutations.

Finally, the genomic coherence values ANI (Average Nucleotide Identity) and Mash genomic distance showed values of 96.7% and 0.02, respectively, between strain E_15.1 and several genomes of reference strains of *A. brasiliensis* (Appendix A), which demonstrates a coherent genetic relationship between the strains.

### 3.4. Identification of Metabolic Pathways and Specialized Metabolites

Three clusters identified as Hybrid NRPS-PKS (non-ribosomal peptide synthetase-polyketide synthase) were detected with a similarity above 60% concerning known BGCs. These corresponded to microperfuranone (region 278.1), terrestric acid (region 345.1), and aspergillicin A (region 389.1). Interestingly, a genomic region with a 100% similarity to the BGC associated with clavaric acid synthesis cluster was observed (Table 2). Moreover, a BLAST inspection of each component of the gene cluster allows us to confirm the ORFs related to the biosynthetic and regulation pathway as well as the transport of the clavaric acid. Analogous biosynthetic gene clusters of clavaric acid in different Aspergillus species were also observed with a synteny analysis (Figure 10).

Of all the genes involved in clavaric acid biosynthesis in strain E_15.1, the one exhibiting the highest identity with the *A. brasiliensis* strain CBS 101740 corresponded to *NTF2* (100%), which is involved in transportation, followed by *Sir2* (99%), which is involved in regulation, and *SqhC* (94%), which plays a key role in the biosynthesis of this metabolite. The *STKc* gene was the one with the lowest percentage of identity (63%) between both strains. The proteins corresponding to *IPP-2* and *Sec14p* were not detected by AntiSMASH within the E_15.1 genome; however, on the set of coding sequences annotated by DIAMOND, both proteins were detected with high homology (100% and 97.3%, respectively) in *A. brasiliensis* CBS 101740 (sequence IDs: g7955.t1 and g7956.t1; Appendix A).

To identify protein families with functions of interest in the genome of strain E_15.1 and the ten compared *Aspergillus* strains, a search for specific metabolic pathways related to potential Plant Growth-Promoting (PGP) attributes was conducted using the KEGG and KofamKOALA orthology databases (Table 3). This analysis revealed the presence of gene copies involved in the biosynthesis of plant growth regulators (AIA, gibberellins), phosphate solubilization, siderophore production, Cytochrome P450, and heat shock proteins (HSP) (Figure 11 and Table 3).

The only gene not found in the genomic context of strain E_15.1 regarding phosphate uptake was *gpmI*, while the rest of the genes were present in a single copy (*phoa* and *PIT*). In the case of *A. pseudoviridinutans*, *A. fumigatus*, and *A. viridinutans*, three copies of PIT and two copies (*phoa*) were found. The *gpmI* gene was only found in a single copy in *A. aculeatus* and *A. carbonarius*. Genes associated with iron uptake were also examined, and three copies of *efeU* were detected in E_15.1, *A. brasiliensis*, *A. carbonarius*, and *A. piperis*, while in the rest of the compared genomes, this gene was found in a single copy.

Two genes involved in an alternative pathway for IAA (indole-3-acetic acid) metabolism (tryptamine; TAM) (*MAO* and *aldH*) were found, with four copies of *aldH* in E_15.1, *A. pseudoviridinutans*, *A. fumigatus*, and *A. viridinutans*, and four copies of *MAO* in *A. vadensis*, *A. brasiliensis*, *A. luchuensis*, and *A. piperis*. Regarding biodegradation and xenobiotic metabolism, three copies of the *ethA* gene were found in E_15.1, *A. aculeatus*, and *A. pseudoviridinutans*. It is worth noting that the highest number of gene copies was found for the *CYP53A1* gene (cytochrome P450), which is attributed to the versatility of these proteins in a wide range of enzymatic reactions.

In relation to the response to abiotic stress, three genes in a single copy were found for all compared genomes (*HSPA1_6_8*, *groES*, *htpG*). Two genes associated with siderophore production (*pvdA* and *argD*) were found in single copies, respectively. Finally, in relation to gibberellin biosynthesis (AG3), a gene (*GGPS1*) was found in two copies in all compared genomes, except for four copies in *A. aculeatus*.

The NMDS analysis revealed two distinct clusters among the different genomes clearly separated in a three-dimensional space (Figure 12). In the first group, the E_15.1 strain was observed close to the strains of *A. brasiliensis*, *A. vadensis*, *A. luchuensis*, and *A. pseudoviridinutans*. In the second group, *A. piperis*, *A. tubingensis*, and *A. carbonarius* were grouped together. Independently and more distant, the strains *A. fumigatus*, *A. viridinutans*, and *A. aculeatus* were observed. These results support the representation of the NMDS, which highlights the genome of *A. aculeatus* as the most distinct species in terms of genomic functional annotations.

### 3.5. In Vitro Assays for PGP Traits and Hydrolytic Activities

Due to the results obtained from the genomic analysis, experiments were performed to confirm the phenotypic characteristics suggested by several interesting traits found in the genome. Through in vitro assays (Figure 13), four positive PGP activities were confirmed in strain E_15.1. A very good P solubilization activity (PVK culture medium) (Figure 13A) was detected for E_1.5, as judged by the broad clear zones around the colonies. Siderophore production using CAS agar medium (Figure 13B) was clearly detected but the halos around the colonies were smaller. The same was true for the gibberellin production using the Candau test (Figure 13C), where the production of fluorescence was observed. The amount of IAA produced by E_15.1 gradually increased with tryptophan concentration (Figure 13G), reaching an IAA production of 14.8 µg/mL with the highest concentration, compared to the medium without tryptophane, where practically no IAA production was observed (0.78 ug/mL). Complementary to this, hydrolytic activity was also detected, specifically for chitinases (Figure 13D), cellulases (Figure 13E), and xylanases (Figure 13F) through the medium culture supplemented with 1% carboxymethylcellulose (CMC), beechwood xylan, or colloidal chitin as the sole carbon source for each assay, respectively. In relation to these results, smaller clear halos could be observed around the colonies when evaluating the activity of chitinases (Figure 13D) and cellulases (Figure 13E) in comparison with more defined and larger halos that were observed in the activity of xylanases (Figure 13F).

### 3.6. Plant Responses to E_15 Interaction in Greenhouse Conditions

We conducted an experiment to evaluate whether E_15.1 can promote plant growth, which is the overall effect of the beneficial properties of a PGPR on the host plant. We evaluated the potential of E_15.1 in enhancing tomato plants’ growth in vivo by applying a suspension of 1 x 10^6^ spores/mL of E_15.1 at the substrate level of tomato plants and assessing their growth after 90 days in greenhouse conditions. The most relevant and statistically significant results were observed in the increased stem longitude, total fresh weight, and total dry mass in the plants inoculated with the E_15.1 strain compared to the positive control and non-inoculated control (Figure 13H).

### 3.7. Effect of Temperature on Xylanase and Cellulase Activities

E_15.1 supernatants were incubated at different temperatures to assess the optimal temperature for xylanase and cellulase activities. The results depicted in Figure 14 show that at 70 °C the optimal temperature for both activities was achieved, with cellulase reaching 2.04 U/mL and xylanase reaching 2.86 U/mL (Figure 14). The cellulase and xylanase activities of the E_15.1 strain at 70 °C were significantly (*p* < 0.05) greater than those at other temperatures. A significant increase in activity was noticeable when the incubation temperature was shifted from 60 °C to 70 °C, clearly demonstrating that these enzymes from the E_15.1 strain exhibit their optimal activity at 70 °C. When assessing the thermostability of these enzymes, it was observed that cellulase activity was lower than that of xylanase (Figure 14) at all temperatures tested but it was more stable since little difference was observed in the activity even after incubation for 1 h at 90 °C (Figure 14), showing a slight decrease after 1 hour at 90 ºC (Figure 14). Instead, xylanase activity decreased as the temperature increased, and it was significantly lower after incubation for 1 hour at 90 °C compared to the other temperatures evaluated (Figure 14). When assessing the residual activity of these enzymes, it was observed that after 1 h of incubation at 90 °C, more than 50% of the xylanase activity was maintained (Figure 14), whereas the cellulase activity was maintained between 70 and 80% (Figure 14). These results confirm that the enzymes tested are robust proteins that could have biotechnological applications.

## 4. Discussion

The Chichonal volcano provides a unique habitat for isolating endophytic fungi from plants, characterized by extreme and fluctuating conditions such as high temperatures, heavy metal concentrations, and low pH values [54]. These polyextremophilic conditions enable the adaptation of organisms with distinctive traits, making the volcano’s crater an ideal site for bioprospecting. Such adapted fungi are valuable for discovering new bioactive compounds with potential applications in agriculture, medicine, and the food industry [55]. Endophytic *Aspergillus* species have been reported for their ability to enhance root and seed development, nutrient absorption, photosynthesis, plant growth, and chlorophyll content [56].

Rapid biochemical tests, such as detecting clear zones around colonies on solid media, are commonly used to assess fungal capabilities [48,57]. This method effectively demonstrated the phosphate solubilization ability of strain E_15.1, underscoring its ecological role in its volcanic habitat. These findings are consistent with studies identifying *Aspergillus* as a dominant phosphate-solubilizing genus in the rhizosphere, particularly *Aspergillus niger*, known for its high solubilization capacity [58]. The phosphorus-solubilizing capability of *Aspergillus* is linked to its production of organic acids, which lower the pH and facilitate solubilization. Acidification occurs through several mechanisms: proton excretion via plasma membrane ATPases, proton exchange during nutrient uptake, organic acid production from metabolic processes, and carbon dioxide production during respiration. These organic acids act as chelating agents, making phosphorus available in the form of orthophosphate [17,59].

Metabolites produced by endophytic fungi play a vital role in soil fertility. Gibberellins (GAs) and indole acetic acid (IAA) are two important growth-regulating metabolites, produced commercially from fungi and used in the agriculture and horticulture industry [51]. Earlier, *A. fumigatus* was reported to promote plant growth, and in vitro culture showed that it also secreted bioactive gibberellins [60]. However, *A. fumigatus* produces several toxins that prevent its use in commercial applications. The intensity of the fluorescence observed with the E_15.1 strain corroborates its positive activity in gibberellin production, a trait directly related to various growth processes, including embryonic development and seed germination induction [61]. The results obtained in the biochemical tests related to plant growth-promoting traits correlate with the results obtained in terms of tomato plant growth promotion in greenhouse conditions. This was reflected in the increased stem and root length, as well as the fresh and dry weight of both shoots and roots in plants inoculated with the E_15.1 strain compared to the not-inoculated control plants. This has been demonstrated by a wide range of growth-promoting microorganisms, primarily belonging to the *Aspergillus* genera, in numerous plant species. For instance, *Pinus koraiensis* plants inoculated with *A. niger*, showed an increase in siderophore production compared to the untreated control [62]. However, it is important to stress that both *A. fumigatus* and *A. niger* can be plant pathogens and produce mycotoxins which are a danger to animal and human consumption, whereas *A. brasiliensis* has never been reported as a dangerous organism.

Overall, the results obtained using molecular tools (Figure 11, Table 2 and Table 3) and culture-based techniques (Figure 13) were consistent. Using the PVK culture medium, the phosphate-solubilizing activity of the E_15.1 strain was determined. Genomic analyses showed the presence of three genes involved in phosphate metabolism. These genes are implicated in the solubilization of inorganic phosphate by modulating its uptake and transport into the cell. It has been reported that passive diffusion of the compound can occur through hydrolysis by the *phoA* gene, which was present in the genome of the E_15.1 strain. Alkaline phosphatases (*phoA*) are among the most common enzymes induced when there are low levels of inorganic phosphate [63].

Through the CAS agar culture medium assay, siderophore production by the E_15.1 strain was observed and was supported by the result of the antiSMASH analysis, which identified a novel siderophore-producing gene cluster in region 345.3 with no matches to the closest-known cluster. Simultaneously, genomic information using the KofamKOALA database for annotation analysis determined that the E_15.1 strain contained a gene associated with iron uptake (*efeU*), which is a high-affinity iron transporter.

Furthermore, the search for genes related to IAA production revealed the presence of *MAO* and *aldH* genes in E_15.1, both involved in the synthesis of this phytohormone. Multiple pathways for IAA synthesis in fungi have been described, most of which use tryptophan as a precursor. An alternative pathway that employs tryptamine as a precursor has been described by Duca and Glick (2020)[64]. This pathway involving tryptamine recruits an amino oxidase enzyme (*MAO*) to convert the primary substrate into indole acetaldehyde, subsequently obtaining IAA through the action of an aldehyde dehydrogenase (*aldH*). Indole-3-acetic acid (IAA) has been reported to be produced by various endophytic fungi such as *A. fumigatus* (EU823312), *Phoma glomerata* (JX111911), *Paecilomyces* sp. (EU823315), *Paecilomyces formosus* (JQ013813), and *Penicillium* sp. (JX111910) [65]. So, E-15.1 seems to have both routes for IAA synthesis.

When the in vitro gibberellin production was assessed, E_15.1 strain exhibited fluorescence intensity, indicative of a positive result. In addition to manual gene search using the KofamKOALA tool, a gene involved in the biosynthesis of gibberellins of the AG3 type (*GGPS1*) was found. This gene belongs to the prenyltransferase family and encodes a protein with geranyl diphosphate (*GGPP*) synthase activity. The enzyme catalyzes the synthesis of *GGPP* from farnesyl diphosphate and isopentenyl diphosphate.

CAZymes are key enzymes for the breakdown and utilization of biopolymers such as cellulose, hemicellulose, pectin, and lignin by fungi allowing them to play a central role in carbon recycling in the biosphere. The fungal CAZome is also closely related to the host preference and adaptation to the fungal lifestyle such as being endophytic [66]. Compared to other *Aspergillus* species, the E_15.1 strain displayed a wide diversity of CAZymes encoding genes, especially GH, CE, and CBM encoding genes, suggesting that the E_15.1 strain may have greater environmental adaptability than other *Aspergillus* species. The analysis of the diversity and abundance of CAZymes encoding genes in these fungi provides insight into how fungi adapt to extreme conditions, such as volcanic environments where nutrients are scarce and difficult to access. In relation to this, the largest number of orthologs related to the metabolism of sugars and sugar nucleotides was represented by K01183 (chitinase), followed by K00698 (chitin synthase) (Figure 5)—results that could be related to the degradation mainly of chitooligosaccharide type substrates of strain E_15.1, which corresponds to the fact that approximately one-third of the predicted CAZy proteins were related to the degradation or chemical modification of chitooligosaccharides. These genes could also play an essential role in fungal cell wall metabolism.

The antiSMASH analyses detected gene clusters related to the biosynthesis of clavaric acid in the E_15.1 strain with 100% identity (Table 2). Synteny analysis among the eleven strains belonging to the *Aspergillus* genus compared in the present study revealed that only five strains presented biosynthetic clusters corresponding to this gene with more than 50% identity (Figure 10).

Although clavaric acid has primarily been associated with Basidiomycetes, its presence as a secondary triterpenoid metabolite has recently been described in *Aspergillus terreus* B12 [67], isolated from a South China sea sponge. The *occ* gene (encoding the squalene oxide cyclase enzyme) has been reported as the first gene in fungi specifically involved in the cyclization of secondary metabolites with a triterpenoid structure, such as the antitumor compound produced by this fungus. One of the peculiarities of this finding is that it opens the door to finding similar occ-like genes in other fungi capable of producing compounds similar to those of the basidiomycete *Hypholoma sublateritium*, where it was initially described by Li et al. (2018) [68]. Clavaric acid has been of interest to researchers due to its antitumor properties, as it is a potent inhibitor of the RAS farnesyltransferase protein, which allows Ras proteins to attach to the cell membrane and send signals. Ras proteins are associated with uncontrolled cell proliferation in a quarter of human tumors, which may have practical relevance in the pharmaceutical industry due to their biological properties in the treatment of different types of cancer [68]. Future studies on clavaric acid biosynthesis in the E_15.1 strain through genetic manipulation may be useful for the development of antitumor medicine.

*Aspergillus* is probably the most known genus for its capacity to produce a broad range of enzymes related to the degradation of plant polysaccharides, such as cellulose, xylan, xyloglucan, and pectin [66]. These enzymes are essential to convert the natural carbon sources of these fungi (mainly plant polymers) into small molecules that can be taken up into the cell and can be widely used in the industry [66].

There have been many reports on thermophilic and mesophilic microorganisms that produce thermostable xylanases. In genus *Aspergillus*, it has been reported that *Aspergillus sydowii* MG 49 produces two xylanases with optimal activity at 60 °C and a stability in the range of 40–70 °C declining sharply around 70 °C [69].

The NMDS analysis in E-15.1 revealed two distinct clusters among the different genomes clearly separated in a three-dimensional space (Figure 12). In the first group, the E_15.1 strain was observed close to the strains of *A. brasiliensis*, *A. vadensis*, *A. luchuensis*, and *A. pseudoviridinutans*. In a second group, *A. piperis*, *A. tubingensis*, and *A. carbonarius* were grouped together. Independently and more distant, the strains *A. fumigatus*, *A. viridinutans*, and *A. aculeatus* were observed. These results support the representation of the NMDS, which highlights the genome of *A. aculeatus* as the most distinct species in terms of genomic functional annotations.

## 5. Conclusions

To date, this is the first study that reports an endophytic and thermophilic *A. brasiliensis* strain. It was isolated in an extreme environment from plants growing in the Chichonal volcano crater and was characterized through molecular tools and culture-based techniques. The fungus exhibited PGP and hydrolytic activity plant growth promotion properties on plants in the tomato variety Saladette in in vitro and greenhouse conditions even outside the optimal date for cultivation. In addition, using genomic information, the presence of genes necessary for iron and phosphate uptake and indole acetic acid and siderophore production, which could contribute to soil biogeochemical processes and improve crop yields, was determined. Thus, strain E_15.1 is an excellent candidate for the evaluation of its contribution as a biocontrol and plant growth-promoting agent in field assays.

## Figures and Tables

**Figure 1 jof-10-00517-f001:**
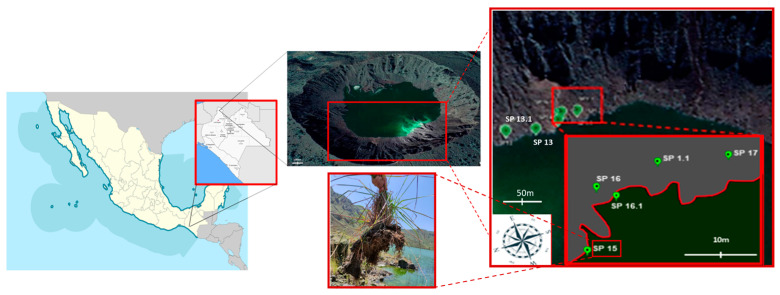
Satellite observation of the Chichonal volcano crater obtained from Google Earth, as well as graphics indicating with higher resolution the sites where samples were collected. SP15, place where plants were found growing of *Andropogon* sp. from where the root sample was taken for the isolation of the endophytic fungus E_15.1.

**Figure 2 jof-10-00517-f002:**
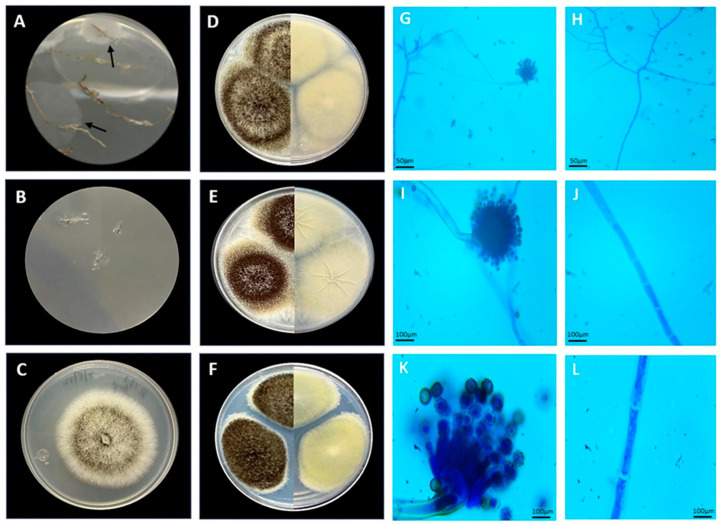
Morphological characteristics of E_15.1 strain isolated of a plant root tissue of *Andropogon* sp. of Chichonal Volcano at 7 days of incubation. (**A**) Roots with mycelial growth at 3 days of cultivation (black arrows), (**B**) Control (disinfection test root imprint showing no mycelial growth), (**C**) Strain E_15.1, Panels (**D**–**F**): left side top view, right sight back view (**D**) Morphology on MEA medium, (**E**) Morphology on CYA medium, (**F**) Morphology on PDA medium, (**G**,**I**,**K**) conidiophore 10, 40, and 100× respectively, (**H**,**J**,**L**) septate hyphae 10, 40, and 100×, respectively.

**Figure 3 jof-10-00517-f003:**
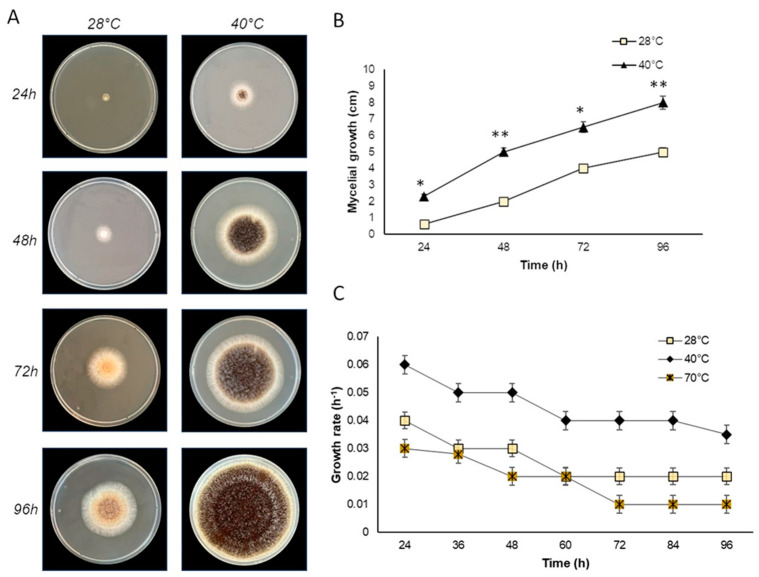
Growth of the endophytic fungal E_15.1 strain isolated from the Chichonal volcano at different temperature conditions. (**A**) Growth of strain E_15.1 on MEA solid medium at different incubation temperatures (28 and 40 °C); (**B**) graph showing the mycelial growth (cm) in the time of strain E_15.1 incubated at different temperatures; (**C**) growth rate of strain E_15.1 in liquid medium incubated at different temperatures. This assay was performed in triplicate and statistical significance was assessed using a Tukey test (* *p* ≤ 0.05, ** *p* ˂ 0.01, statistical significance indicated by asterisks).

**Figure 4 jof-10-00517-f004:**
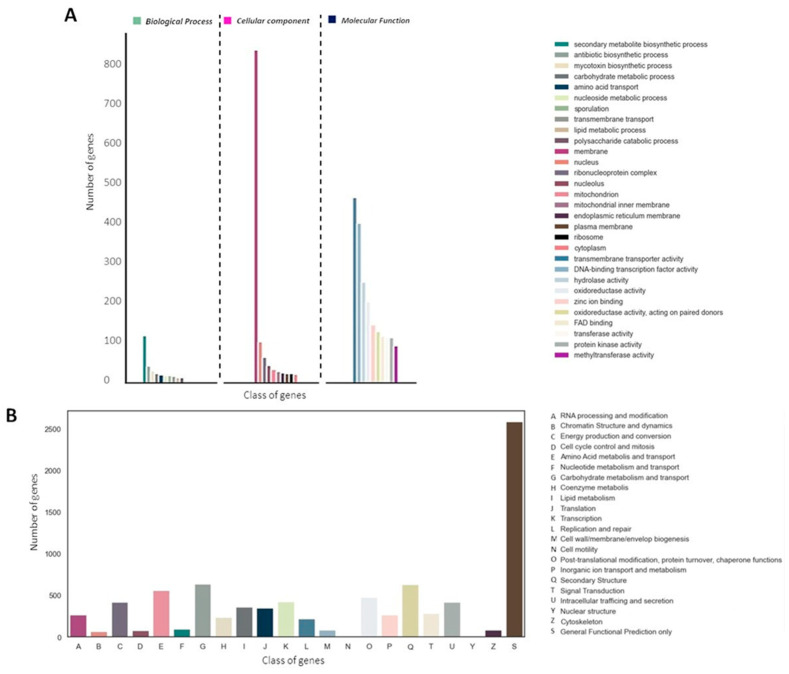
Annotation and functional classification of E_15.1 strain. (**A**) Gene Ontology (GO) classification, (**B**) histogram of Cluster of Orthologs Groups (COG).

**Figure 5 jof-10-00517-f005:**
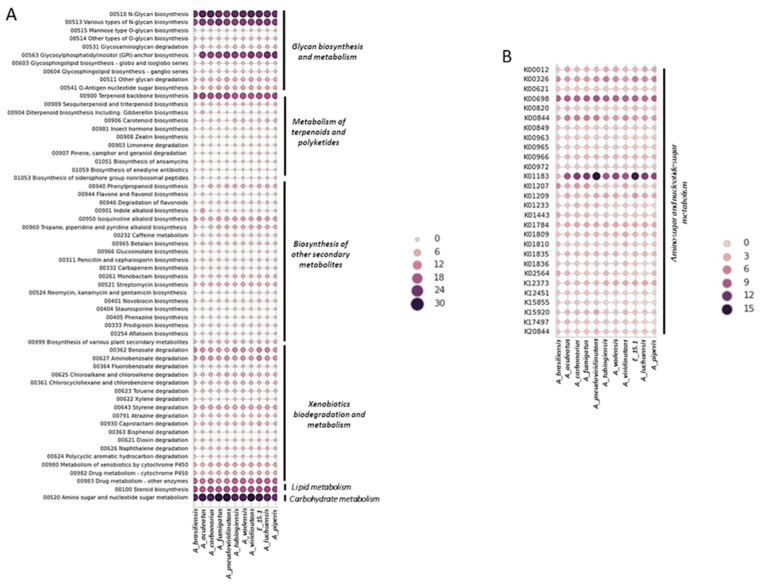
Functional annotation by KEGG of the proteins predicted by AUGUSTUS from the fungal genome E_15.1 and 10 strains belonging to the *Aspergillus* genus. (**A**) Main pathways of secondary metabolism, xenobiotic degradation, and lipid and carbohydrate metabolism. (**B**) Orthologs related to amino sugars metabolism.

**Figure 6 jof-10-00517-f006:**
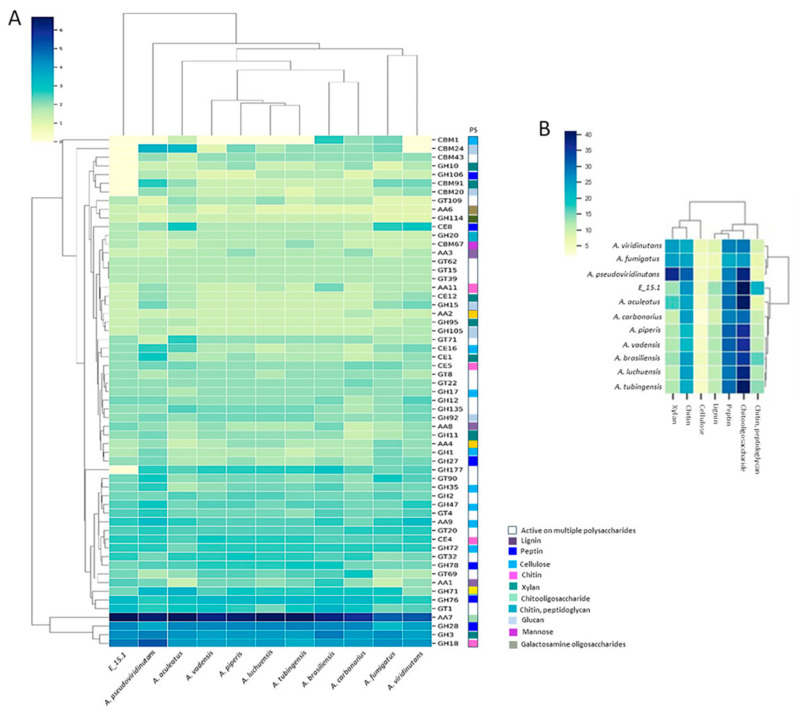
Functional annotation of carbohydrate-active enzymes (CAZymes) encoded by the E_15.1 genome in comparison to CAZymes identified in the genomes of selected fungi. (**A**) Number of genes per CAZy family related to plant biomass degradation. (**B**) Abundance of coding sequences for CAZymes targeting different substrates (PS) in related fungal genomes. Enzyme families are represented by their classes (GH: glycoside hydrolases, GT: glycosyltransferases, PL: polysaccharide lyases, CE: carbohydrate esterase, and CBM: chitin-binding modules), and the family numbers are based on HMM predictions from the carbohydrate-active enzyme database. The bars of different colors represent the different substrates (PS) on which the different classes of CAZymes act. The abundance levels of different enzymes within a family are depicted using a color scale, from the lowest (yellow) to the highest occurrences (dark blue) per species.

**Figure 7 jof-10-00517-f007:**
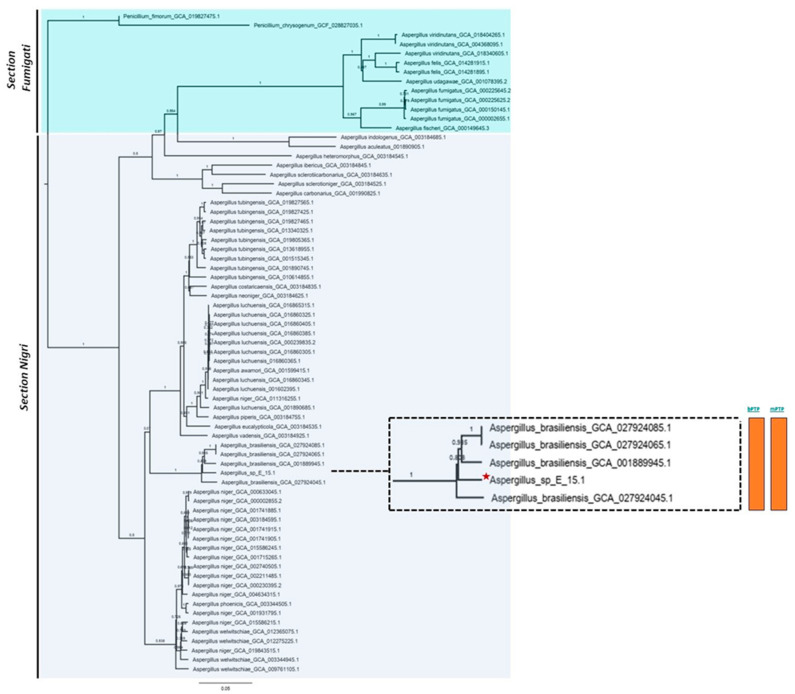
Phylogenomic analysis of the *Aspergillus* genus based on mutational distances using the JolyTree Program (v.1.1b.191021ac) with RefSeq category genomes. The orange color bars show that species delimitation tests in both algorithms (bPTP and mPTP) restrict strain E_15.1 to a coalescence rate with all *A. brasiliensis* species.

**Figure 8 jof-10-00517-f008:**
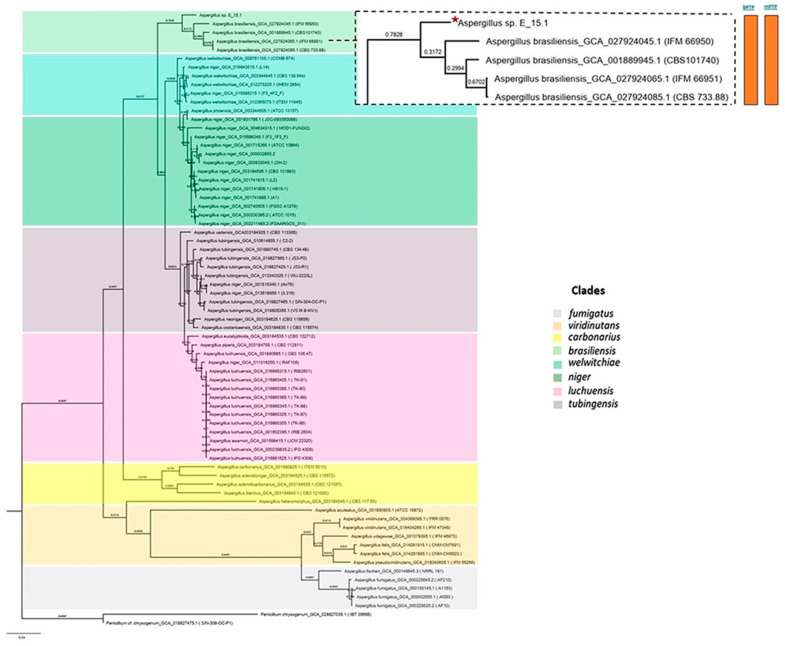
Phylogenetic orthology with the OrthoFinder Program (v2.5.4). The orange color bars show that species delimitation tests in both algorithms (bPTP and mPTP) restrict strain E_15.1 to a coalescence rate with all *A. brasiliensis* species.

**Figure 9 jof-10-00517-f009:**
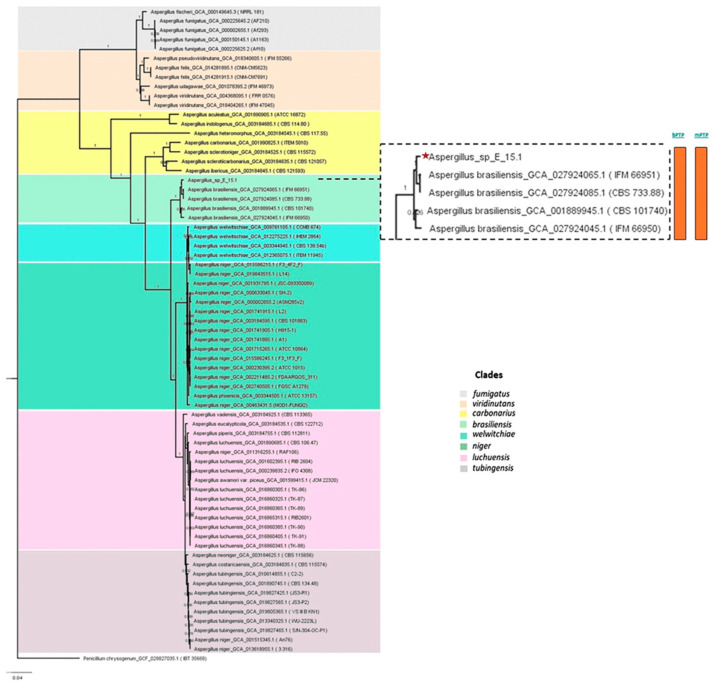
Phylogenetic analysis using UFCG (Universal Fungi Core Genes) Program (v1.0.3). The orange color bars show that species delimitation tests in both algorithms (bPTP and mPTP) restrict strain E_15.1 to a coalescence rate with all *A. brasiliensis* species.

**Figure 10 jof-10-00517-f010:**
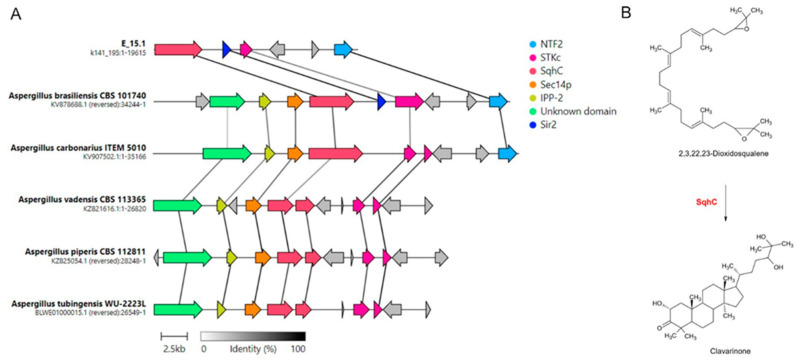
Synteny analysis of clavaric acid biosynthetic gene clusters identified in different *Aspergillus* species. (**A**) Representation of the best link (highest protein sequence similarity) for each gene; (**B**) cyclization reaction of the terpene 2, 3, 22, 23 dioxidosqualene by the action of the enzyme *SqhC* (squalene cyclase) to obtain clavarinone, a direct precursor of clavaric acid. *STKc*: serine/threonine protein kinase; *NTF2*: nuclear transport factor 2; *Sec14p*: phosphatidylinositol transfer protein; *IPP-2*: protein phosphatase 2 (phosphoprotein) inhibitor; *Sir2*: NAD-dependent protein deacetylase.

**Figure 11 jof-10-00517-f011:**
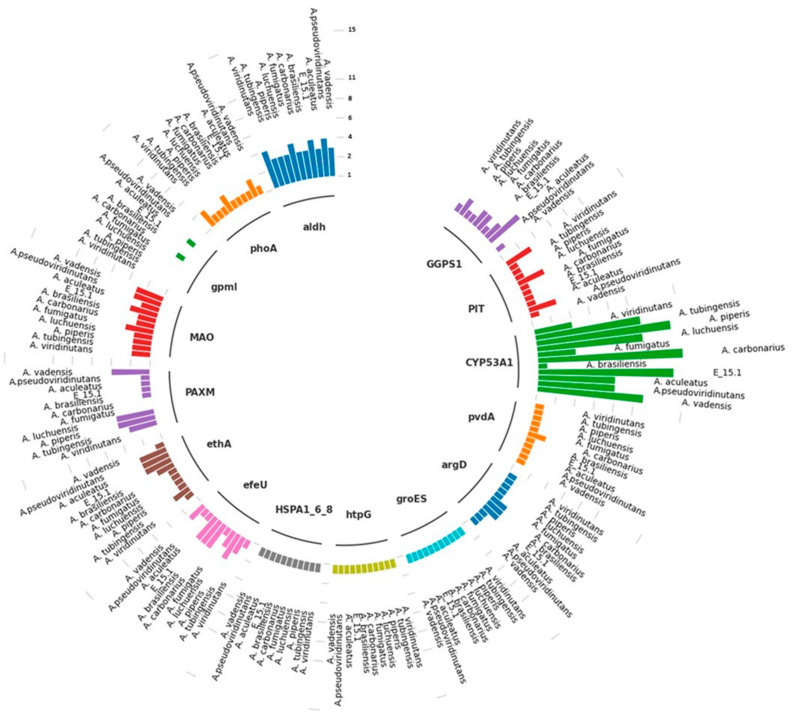
Different copies of genes involved in specific metabolic pathways associated with plant growth promotion, xenobiotics biodegradation and metabolism, and abiotic stress response traits found in the genome of E_15.1 strain and ten strains of the genus *Aspergillus* compared through KEEG orthology and KofamKOALa databases. The numerical scale indicates the number of copies in the genome.

**Figure 12 jof-10-00517-f012:**
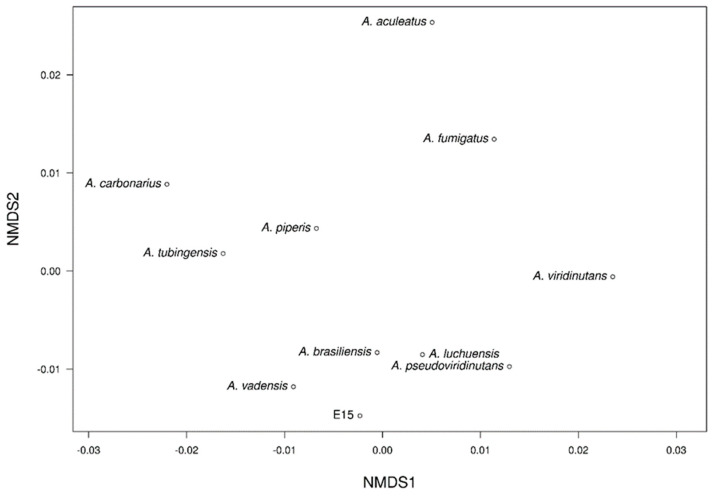
Non-metric multi-dimensional scaling (NMDS) clustering based on Euclidean distance of eleven *Aspergillus* genomes based on similarity of functional groups according to Pfam, COG, KEGG, CAZy, and antiSMASH annotations.

**Figure 13 jof-10-00517-f013:**
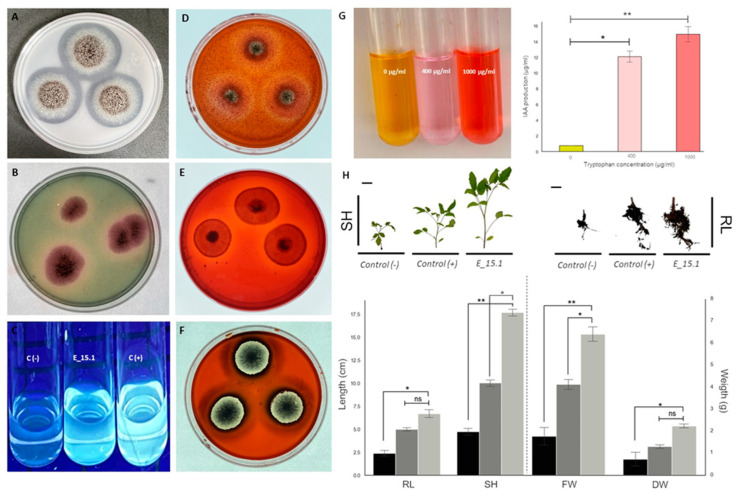
In vitro assays showing PGP and hydrolytic activity traits of *Aspergillus brasiliensis* strain E_15.1. (**A**) Phosphate solubilization test in PVK medium; (**B**) production of siderophores; (**C**) gibberellin production, C (−) indicates the negative control using non-inoculated medium; C (+) indicates the positive control using 2 µg/mL AG3; (**D**) chitinase activity; (**E**) cellulase activity; (**F**) xylanase activity, (**G**) IAA production, right—visual test color with different trp concentrations, left—amount of IAA according to the calculations using the standard curve; (**H**) effect of inoculation of *A. brasiliensis* E_15.1 on Saladette variety tomato plants. Plants inoculated with 1 mL of sterile water were used as negative controls (black bars); for comparison, 1 × 10^6^ spores/mL of *Trichoderma atroviridae* were inoculated to the plants (dark gray bars); finally, 1 × 10^6^ spores/mL of E_15.1 inoculated plants (light gray bars) were assessed for growth promotion traits. The following metrics were recorded after 90 days: total fresh weight (FW), total dry mass (DW), root length (RL), and shoot height (SH). This assay was performed in triplicate and statistical significance was assessed using a Tukey test (* *p* ≤ 0.05, ** *p* ˂ 0.01, statistical significance indicated by asterisks). Ns: non-significant. Scale bars = 1 cm.

**Figure 14 jof-10-00517-f014:**
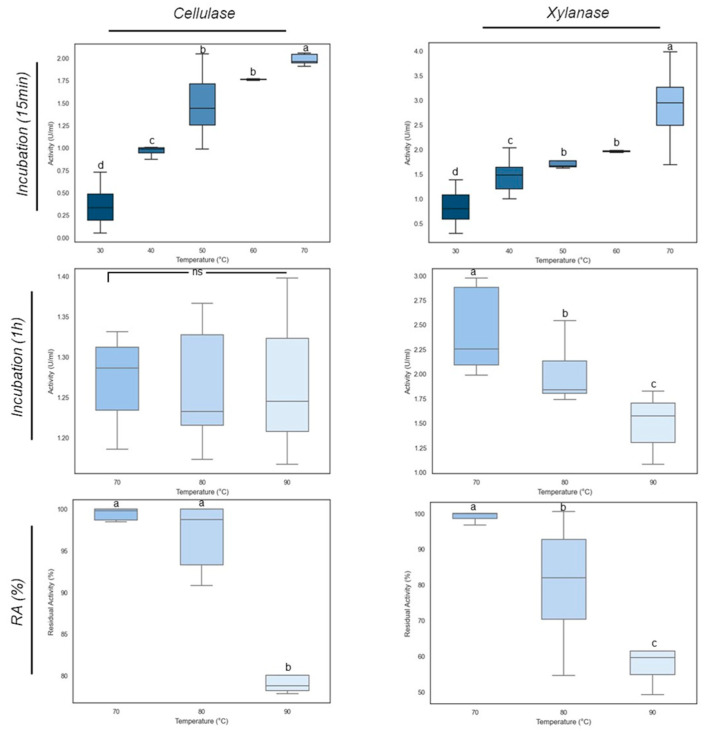
Effect of temperature on xylanase and cellulase activities of E_15.1 strain. The two upper panels show the optimal temperature for each kind of enzyme. The four lower panels show the thermostability of the cellulase and xylanase activities. The horizontal bars within each box represent the mean of the triplicates while the lines outside the boxes show the standard deviation. Different letters indicate significant differences determined by one-way ANOVA and Tukey’s multiple mean comparisons when *p* ≤ 0.05. Ns: non-significant. RA: remaining activity after incubation one hour at the depicted temperature.

**Table 1 jof-10-00517-t001:** Summary of several main features for E_15.1 strain compared to ten sequenced *Aspergillus* genomes.

Species	Strain	Accession Number	Genome Size (Mb)	Content G + C (%)	Predicted Proteins
*A. brasiliensis*	CBS 101740	GCA_001889945.1	35.8	50.25	10,120
*A. brasiliensis*	E_15.1	JBBEEP010000000	36.9	49.4	10,992
*A. piperis*	CBS 112811	GCA_003184755.1	35.2	48.99	9702
*A. viridinutans*	IFM 47045	GCA_018404265.1	34.9	45.41	9314
*A. luchuensis*	IFO 4308	GCA_016861625.1	37.2	48.82	9954
*A. carbonarius*	ITEM 5010	GCA_001990825.1	36.1	48.82	10,071
*A. aculeatus*	ATCC 16872	GCA_001890905.1	35.4	50.52	10,196
*A. pseudoviridinutans*	IFM 55266	GCA_018340605.1	33.3	49.33	10,005
*A. tubingensis*	WU 2223-L	GCA_013340325.1	35.0	49.32	9667
*A. vadensis*	CBS 113365	GCA_003184925.1	35.6	49.18	9642
*A. fumigatus*	AF 293	GCA_000002655.1	29.3	48.82	8681

**Table 2 jof-10-00517-t002:** Secondary metabolites gene sequences identified in E_15.1 strain.

Region	Type	From	To	Most Similar Known Cluster	Similarity
26.1	T1PKS	1	52,794	3’-methoxy-1,2-dehydropenicillide	26%
489.1	terpene	55,448	86,734	-	-
332.1	terpene	65,589	97,901	oryzine A	12%
216.1	terpene	1	19,615	clavaric acid	100%
405.1	indole	1540	32,820	-	-
345.3	NI-siderophore	627,350	645,079	-	-
418.1	NRPS	1	61,789	-	-
278.1	Hybrid NRPS-PKS	13,579	156,906	microperfuranone	66%
345.1	Hybrid NRPS-PKS	128,207	200,446	terrestric acid	62%
377.1	T1PKS	17,129	82,533	yanuthone D	70%
389.1	Hybrid NRPS-PKS	7514	142,103	aspergillicin A	66%
487.2	Fungal-RiPP-like	138,687	229,455	terreazepine	7%
52.1	NRPS-like	72,815	136,204	notoamide A	11%
158.2	NRPS	262,537	331,184	azanigerone A	13%

**Table 3 jof-10-00517-t003:** Strain E_15.1 genes associated with plant growth promotion, xenobiotics biodegradation and metabolism, and abiotic stress response traits discussed in this study.

Associated Function	Annotation Entry (KO)	Annotation Pfam	Gene	Product Name
Phosphate uptake	K01077	PF00245	*phoA*	alkaline phosphatase
	K15633	PF00245	*gpmI*	2,3-bisphosphoglycerate-independent phosphoglycerate mutase
	K14640	PF01384	*PIT*	solute carrier family 20 (sodium-dependent phosphate transporter)
IAA	K00128	PF00171	*aldH*	Aldehyde dehydrogenase family
	K00274	PF00743	*MAO*	monoamine oxidase
	K01667	PF00155	*tnaA*	tryptophanase
	K18386	PF01593	*PAXM*	FAD dependent monooxygenase
Xenobiotics biodegradation and metabolism	K10215	PF00743	*ethA*	monooxygenase
Cytochrome P450	K07824	PF00067	*CYP53A1*	benzoate 4-monooxygenase
Abiotic stress response	K03283	PF00012	*HSPA1_6_8*	heat shock 70kDa protein 1/6/8
	K04078	PF00166	*groES*	chaperonin GroES
	K04079	PF00183	*htpG*	molecular chaperone HtpG
Iron uptake	K07243	PF03239	*efeU*	high-affinity iron transporter
Siderophore	K10531	PF13434	*pvdA*	L-ornithine N5-monooxygenase
	K00818	PF00155	*argD*	acetylornithine aminotransferase
Biosynthesis of gibberellins (GAs)	K00804	PF03936	*GGPS1*	geranylgeranyl diphosphate synthase, type III

## Data Availability

The raw sequencing data and annotated assembled genomes have been deposited at DDBJ/ENA/GenBank under the BioProject accession JBBEEP000000000. The version described in this paper is version JBBEEP010000000.

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
