# Peer review of "Aspergillus brasiliensis E_15.1: A Novel Thermophilic Endophyte from a Volcanic Crater Unveiled through Comprehensive Genome-Wide, Phenotypic Analysis, and Plant Growth-Promoting Trails"

_jof, 2024, doi:10.3390/jof10080517_

Round 1

Reviewer 1 Report

This is interesting research to discover a novel strain of Aspergillus brasiliensis (A. brasiliensis E15.1). Then the author did a series of analysis to this new stain including

 phylogenomic analysis, functional traits identification, hydrolytic enzymatic activity test, plant responses, as well as statistical analysis. Their results showed this A. brasiliensis E15.1 has a high potential for industrial application due to its thermostability and can promote plant growth at high temperatures. Also, this strain constitutes an interesting source of terpenoids with pharmacological activity.  The topic and results of this manuscript show it is an interesting paper and worth acceptance. However, the paper's presentation could be more engaging and visually appealing. The language requires further revision. Therefore, I prefer to give a major revision to this manuscript. I would be happy to reconsider it after these revisions have been made.

1.     This research deserves a more appealing title. Please try to find an attractive title.

2.     Similarly, the description of the abstract is too plain. In line 27, “The average temperature of the collection site was around 60 °C and the soil pH was moderately acidic.” This sentence was suggested to be revised as “This strain is capable of surviving in soil with an ambient temperature of 60 °C and a pH of neutral acidity, which have a high thermostability in industrial application.”

3.     “The complete genome of A. brasiliensis E_15.1 was sequenced and assembled in about 37 Mb of genomic DNA.” This sentence was suggested to be deleted.

4.     “Among the functional traits identified was the degradation of chitin, chitooligosaccharides, xylan, and cellulose, as well as the biosynthesis of clavaric acid, a triterpene with antitumor activity due to its ability to inhibit Farnesyl-protein transferase.”  This sentence was suggested to be revised as “A. brasiliensis E_15.1 was found to degrade chitin, chitooligosaccharides, xylan, and cellulose, and was also able to biosynthesize clavulanic acid thus having antitumor activity.”

5.     “Our findings suggest that the enzymatic repertoire…….”  This sentence was suggested to be revised as “Enzymatic repertoire of A. brasiliensis E_15.1 suggest that……”

6.     The abstract was strongly suggested to be rewritten.

7.     In lines 88-101, could you please restate this paragraph in one or two sentences and include it in the subsequent paragraph?

8.     In lines 121-125, this sentence is too long to follow.

9.     “A lake has formed inside the crater with huge temperature and pH, variations from the volcano vent to other more distant places within the crater.” This sentence was suggested to be deleted.

10.  In the methodology section, please briefly describe the method or experimental procedures in a few sentences.   The description is excessively detailed and therefore difficult to focus on. It is not necessary to present every action performed. Less is more. I hope you can fully revise it.

11.  “The microscopic characterization with lactophenol staining of mycelium grown in MEA culture medium after five days of incubation at 28 °C, the presence of conidiophores with phialides and conidia with light green pigmentation could be observed”. If you mentioned this culture condition in Method, do not repeat it and try to simplify your sentence.

12.  In lines 506-509, 524-527, 534-537can you rewrite these sentences to make them easier to follow?

13.  The manuscript contains numerous lengthy sentences, interspersed with a lack of logical connectives, which impede comprehension of the author's intended meaning. Even though I know you do beautiful work.

Author Response

Answers to Reviewer1
In green: answers to the reviewer
In blue actual changes made to the manuscript (and in the manuscript itself)
In red, deletions to the text

(see cover letter for the colors)

This is interesting research to discover a novel strain of Aspergillus brasiliensis (A. brasiliensis E15.1). Then the author did a series of analysis to this new stain including phylogenomic analysis, functional traits identification, hydrolytic enzymatic activity test, plant responses, as well as statistical analysis. Their results showed this A. brasiliensis E15.1 has a high potential for industrial application due to its thermostability and can promote plant growth at high temperatures. Also, this strain constitutes an interesting source of terpenoids with pharmacological activity. The topic and results of this manuscript show it is an interesting paper and worth acceptance. However, the paper's presentation could be more engaging and visually appealing. The language requires further revision. Therefore, I prefer to give a major revision to this manuscript. I would be happy to reconsider it after these revisions have been made.

Detail comments
1.     This research deserves a more appealing title. Please try to find an attractive title.
Pensar título
2.     Similarly, the description of the abstract is too plain. In line 27, “The average temperature of the collection site was around 60 °C and the soil pH was moderately acidic.” This sentence was suggested to be revised as “This strain is capable of surviving in soil with an ambient temperature of 60 °C and a pH of neutral acidity, which have a high thermostability in industrial application.”

Thank you for the suggestion. We changed the sentence to “This strain was capable of surviving in soil with a temperature of 60 °C and a pH of neutral acidity, which preluded a high thermostability and a potential in industrial application”. Just to make it clear the soil was at 60 ºC, but not the air above it, so not the whole environment of the association between the fungus and the plant was at the same temperature (which was significantly lower, so the aerial part of the plant was around 40 ºC.)

3. “The complete genome of A. brasiliensis E_15.1 was sequenced and assembled in about 37 Mb of genomic DNA.” This sentence was suggested to be deleted.

We believe that this sentence should remain because it supports the next paragraphs (the phylogenomic analysis, and all the other whole genome tests that were performed, which, without the whole genome sequence could have not been assessed)

4.     “Among the functional traits identified was the degradation of chitin, chitooligosaccharides, xylan, and cellulose, as well as the biosynthesis of clavaric acid, a triterpene with antitumor activity due to its ability to inhibit Farnesyl-protein transferase.”  This sentence was suggested to be revised as “A. brasiliensis E_15.1 was found to degrade chitin, chitooligosaccharides, xylan, and cellulose, and was also able to biosynthesize clavulanic acid thus having antitumor activity.”

Thank you for the suggestion. The sentence was divided in two: 
“A. brasiliensis E_15.1 was found to degrade chitin, chitooligosaccharides, xylan, and cellulose. The genes to biosynthesize clavaric acid (a triterpene with antitumor activity) were found, thus probably having antitumor activity

We did not measure the production of clavaric acid, we only found the gene cluster conserved as in other Aspergilli. So, this indicates a potential to produce this compound, but additional research must be done to prove this hypothesis

5.     “Our findings suggest that the enzymatic repertoire…….”  This sentence was suggested to be revised as “Enzymatic repertoire of A. brasiliensis E_15.1 suggest that……”

Thank you, the suggestion has been made in the text.

6.     The abstract was strongly suggested to be rewritten.

We have improved the abstract with your suggestions and other changes in the writing

7.     In lines 88-101, could you please restate this paragraph in one or two sentences and include it in the subsequent paragraph?

This paragraph has been simplified and split in two separate paragraphs

8.     In lines 121-125, this sentence is too long to follow.

Thank you. Indeed, it was too long and difficult to follow. Now it has been rewritten in several sentences

9.     “A lake has formed inside the crater with huge temperature and pH, variations from the volcano vent to other more distant places within the crater.” This sentence was suggested to be deleted.

We believe that this sentence should remain since it describes the environment from which the fungus was isolated. Precisely the variations in temperature, pH, heavy metal content along the lake could be the driving forces that select novel strains that adapted to these highly unstable conditions. It should be noted that the roots of the plant from which this strain was isolated were inside the water of the lake, close to the shore

10.  In the methodology section, please briefly describe the method or experimental procedures in a few sentences.   The description is excessively detailed and therefore difficult to focus on. It is not necessary to present every action performed. Less is more. I hope you can fully revise it.

Thank you for your suggestion, we have made the materials and Methods section shorter by deleting nonrelevant information (in red in the manuscript). Nevertheless, some details remain in this section, especially those regarding the enzymatic assay, since this varies among different research groups. Also, the bioinformatic section must be detailed since the other reviewer proposes that this strain is a novel species, when all the tests indicate that is A. brasiliensis.

11.  “The microscopic characterization with lactophenol staining of mycelium grown in MEA culture medium after five days of incubation at 28 °C, the presence of conidiophores with phialides and conidia with light green pigmentation could be observed”. If you mentioned this culture condition in Method, do not repeat it and try to simplify your sentence.

The sentences are not the same. The one you mention is the result of the observation. In the methods section it only states which method was used

12.  In lines 506-509, 524-527, 534-537can you rewrite these sentences to make them easier to follow?

These sentences have been simplified by deleting information that is present in Table 3, now we think is easier to follow.

13.  The manuscript contains numerous lengthy sentences, interspersed with a lack of logical connectives, which impede comprehension of the author's intended meaning. Even though I know you do beautiful work.

Thank you. We have looked carefully to the style of the document writing and tried to simplify it, so it follows a more logical reading.

Reviewer 2

The work was carried out at a good methodological level and represents a comprehensive study, including both a genome-wide analysis of the isolated Aspergillus brasiliensis isolate and various biochemical tests. A test was also carried out for endophytic activity during the infestation of tomato plants in a greenhouse. The work is well presented, the amount of research is significant, this makes the work look holistic.
The authors compared the genome of the studied Aspergillus with the closest species and saw strong differences (Table 1.). However, for some reason, the new strain was not isolated as a new species, or Aspergillus sp., but was named Aspergillus brasiliensis, which raises questions. Since the strain isolated by the authors has significant differences both in the genomic sequence and in physiology associated with thermophilicity. Thus, A. brasiliensis CBS 101740 GCA_001889945.1 has a genome size of 35.8 Mb, and A. brasiliensis E_15.1 JBBEEP010000000 has a genome size of 36.9 Mb. The fact that the authors are dealing with a new species of Aspergillus is also indicated by the data in Figures 6, 7, and 10. In this regard, I would like to receive a detailed answer from the authors, based on what criteria they decided not to introduce a new species of Aspergillus, but to attribute the resulting isolate to Aspergillus brasiliensis. Otherwise, there are no comments on the work; it can be accepted for publication after minor corrections given below.

You are right. At first, we thought it was a new species since the numbers that you mention could be relevant. Also, the GC% content is a little bit different from the reference strain CBS 101740 GCA_001889945.1. The number of predicted proteins is also higher in E_15.1 than in the reference sequence. However interspecies GC values differences are expected to be between 1 and 2 %. In our case the value is only 0.74%. Regarding the genome size, Aspergilli show a big variability that goes between 20 to 40 Mb (approximately) and E_15.1 has only a 1.1 % difference with the reference strain.

Previous phylogenetic analysis with 13 A. brasiliensis strains, performed with three concatenated molecular markers (ITS, -tubulin and calmodulin), placed E-15.1 in the A. brasiliensis clade (data not shown).
However, having the whole genome sequence allowed us to perform detailed state-of-the -art analyses that undoubtedly showed that this is a strain of A. brasiliensis. Probably the differences stated above are due to the stress conditions in which it was isolated (gene duplication, for example).
Analyses preformed using robust bioinformatic tools such as Orthofinder, Jolytree and UFCG to infer phylogenetic trees, resulted that E-15.1 groups with four complete genomes of A. brasiliensis strains deposited in the GenBank, forming a monophyletic group in all the analyzed phylogenies. Besides, speciation tests using bPTP and mPTP also showed that this strain is A. brasiliensis.
Additionally, genomic coherence measurements such as the global genomic relation index (OGRI) showed comparative and representative values for A. barslilensis in all cases. Values of ANI (Average Nucleotide Identity) and Mash genomic distances showed values above 95%, which is the threshold to determine if a strain belongs to that species. With E.15.1 we obtained values of 96.7% and 0.02 respectively, comparing strain E_15.1 and the genome of reference strain GCA001889945.1. Although this is mentioned in the manuscript, a table with this data has been included as supplementary material.
Still not fully convinced we submitted the E.1.5 genome to a k-mer analysis using the FOCUS software; this tool allows to detect alleles and mutations between species and in our case the obtained value was 90.7 % of identical k-mers between A. brasiliensis E_15.1 and A. brasiliensis IFM_66951, indicating that E_15.1 is indeed a A. brasiliensis species.

Detail comments
Lines 48-50
In the abstract you say that "The average temperature of the collection site was around 60 °C". In addition, you collected plant samples (Andropogon  sp. )from which the thermophilic fungus A. brasiliensis was extracted. Therefore, the host plants were exposed to the same thermophilic conditions, if not even more extreme. In this regard, it is not clear why you say that “only a limited number of fungal species are recognized as genuinely thermophilic”. What about the host organism? Why can't host plants also be called genuinely thermophilic? And the fact that they would not become thermophilic without an endophytic fungus also cannot exclude them from thermophilic, since endosymbiosis is one of the adaptation mechanisms.

You are right. It is very possible that the plant is also thermophilic. Unfortunately, we were unable to breed the plant in the laboratory, so we could not test its thermal preferences with and without the endophyte.
The reference to genuine thermophilic fungi is quoted according to the consulted literature. Thermophilic organisms are those that grow better at higher temperatures than related species. In this case, E-15.1 grew optimally at 40 ºC, while most Aspergilli are reported to grow optimally around 28 ºC. However, this strain provided tolerance to heat stress in a different plant from which it was isolated (S. lycopersicum) suggesting a broad host interaction with plants. Certainly, other plants species (thermophilic or not) should be tested to determine the range of interaction with this strain. Just as an example, Trichoderma species are not too specific regarding the plant species they can colonize.

Line 50.
Incorrect citation of a literary source in connection with The upper-temperature threshold for eukaryotes. The primary source is Tansey MR, Brock TD (1972) The upper temperature limit for eukaryotic organisms. Proc Natl Acad Sci U S A 69:2426–2428. And the indicated literary source itself cites the article Tansey MR, Brock TD (1972)

The reference refers to fungi, not all eukaryotes. As they are mentioned in a previous sentence, we will include the reference that you suggest. Thank you

Line 73
Incorrect citation number 7. (Ali, A.H.; Abdelrahman, M.; Radwan, U.; El-Zayat, S.; El-Sayed, M.A. Effect of Thermomyces Fungal Endophyte Isolated from 834 Extreme Hot Desert-Adapted Plant on Heat Stress Tolerance of Cucumber. Applied Soil Ecology 2018, 124, 155–162, 835 doi:10.1016/j.apsoil.2017.11.004.). This article does not describe Curvularia protuberate.

You are right, the title of the paper was wrong It has now been corrected:

Ali, A.H.; Abdelrahman, M.; Radwan, U.; El-Zayat, S.; El-Sayed, M.A. Desert plant-fungal endophytic association: the beneficial aspects to their hosts. Biological Forum-An International Journal 2018, 10, 138–145.

Line 75
Incorrect citation of a literary source, The primary source is - Redman RS, Sheehan KB, Stout RG, Rodriguez RJ, Henson JM (2002) Thermotolerance generated by plant/fungal symbiosis. Science 298:1581

This has been also corrected 

[8] Redman, R.S.; Sheehan, K.B.; Stout, R.G.; Rodriguez, R.J.; Henson, J.M. Thermotolerance Generated by Plant/Fungal Symbiosis. Science 2002, 298, 1581, doi:10.1126/science.1072191.

Lines 111-113
Reword the sentence, in the current version you get that siderophores are a heat-stable enzyme, like phytases. And this is wrong, since siderophores are low-molecular compounds.

Thank you, you are rigth. Now ir reads:

…” these fungi can produce siderophores and heat-stable enzymes, such as phytases,

Move the sampling map (supplementary file 1) into the body of the article. Information related to the local location of an object is of interest to readers, and supplementary information is less frequently viewed than the main text of the article. Improve the design of the Figure. In its current form, its lower left part is empty.

The Figure has been included in the main text and modified to better use the space for it.

Figure 1
This figure is oversaturated with information, too many numbers, from A to O. This makes some information difficult to perceive. In particular, in Figure 1.O it is not clear which part of Figure corresponds to which temperature. Divide this Figure into two Figures, in the first one leave A-L, to the second Figure move M-O. This will make the material easier to understand.

Thank you for your suggestion, we have divided the figure in two separate figures (2 and 3), carefully revising the figure order and their mention in the text.

Line 415
In the current version of submission, the source you specified (Supplementary Table S2) is not attached. Only resource available: Dataset S1. NCBI non-redundant database annotation of E_15.1 predicted proteins

We have now made sure that all supplementary material is in the file. Thank you for your comments

Reviewer 2 Report

The work was carried out at a good methodological level and represents a comprehensive study, including both a genome-wide analysis of the isolated Aspergillus brasiliensis isolate and various biochemical tests. A test was also carried out for endophytic activity during the infestation of tomato plants in a greenhouse. The work is well presented, the amount of research is significant, this makes the work look holistic.

The authors compared the genome of the studied Aspergillus with the closest species, and saw strong differences (Table 1.). However, for some reason, the new strain was not isolated as a new species, or Aspergillus sp, but was named Aspergillus brasiliensis, which raises questions. Since the strain isolated by the authors has significant differences both in the genomic sequence and in physiology associated with thermophilicity. Thus, A. brasiliensis CBS 101740 GCA_001889945.1 has a genome size of 35.8 Mb, and A. brasiliensis E_15.1 JBBEEP010000000 has a genome size of 36.9 Mb. The fact that the authors are dealing with a new species of Aspergillus is also indicated by the data in Figures 6, 7, and 10. In this regard, I would like to receive a detailed answer from the authors, based on what criteria they decided not to introduce a new species of Aspergillus, but to attribute the resulting isolate to Aspergillus brasiliensis. Otherwise, there are no comments on the work; it can be accepted for publication after minor corrections given below.

Lines 48-50

In the abstract you say that "The average temperature of the collection site was around 60 °C". In addition, you collected plant samples (Andropogon  sp. )from which the thermophilic fungus A. brasiliensis was extracted. Therefore, the host plants were exposed to the same thermophilic conditions, if not even more extreme. In this regard, it is not clear why you say that “only a limited number of fungal species are recognized as genuinely thermophilic”. What about the host organism? Why can't host plants also be called genuinely thermophilic? And the fact that they would not become thermophilic without an endophytic fungus also cannot exclude them from thermophilic, since endosymbiosis is one of the adaptation mechanisms.

Line  50

Incorrect citation of a literary source in connection with The upper-temperature threshold for eukaryotes. The primary source is Tansey MR, Brock TD (1972) The upper temperature limit for eukaryotic organisms. Proc Natl Acad Sci U S A 69:2426–2428. And the indicated literary source itself cites the article Tansey MR, Brock TD (1972)

Line 73

Incorrect citation number 7. (Ali, A.H.; Abdelrahman, M.; Radwan, U.; El-Zayat, S.; El-Sayed, M.A. Effect of Thermomyces Fungal Endophyte Isolated from 834 Extreme Hot Desert-Adapted Plant on Heat Stress Tolerance of Cucumber. Applied Soil Ecology 2018, 124, 155–162, 835 doi:10.1016/j.apsoil.2017.11.004.). This article does not describe Curvularia protuberate.

Line 75

Incorrect citation of a literary source, The primary source is - Redman RS, Sheehan KB, Stout RG, Rodriguez RJ, Henson JM (2002) Thermotolerance generated by plant/fungal symbiosis. Science 298:1581

Lines 111-113

Reword the sentence, in the current version you get that siderophores are a heat-stable enzyme, like phytases. And this is wrong, since siderophores are low-molecular compounds.

Move the sampling map (supplementary file 1) into the body of the article. Information related to the local location of an object is of interest to readers, and supplementary information is less frequently viewed than the main text of the article. Improve the design of the Figure. In its current form, its lower left part is empty.

Figure 1

This figure is oversaturated with information, too many numbers, from A to O. This makes some information difficult to perceive. In particular, in Figure 1.O it is not clear which part of Figure  corresponds to which temperature. Divide this Figure into two Figures, in the first one leave A-L, to the second Figure move M-O. This will make the material easier to understand.

Line 415

In the current version of submission, the source you specified (Supplementary Table S2) is not attached. Only resource available: Dataset S1. NCBI non-redundant database annotation of E_15.1 predicted proteins

Author Response

Reviewer 2

The work was carried out at a good methodological level and represents a comprehensive study, including both a genome-wide analysis of the isolated Aspergillus brasiliensis isolate and various biochemical tests. A test was also carried out for endophytic activity during the infestation of tomato plants in a greenhouse. The work is well presented, the amount of research is significant, this makes the work look holistic.

The authors compared the genome of the studied Aspergillus with the closest species and saw strong differences (Table 1.). However, for some reason, the new strain was not isolated as a new species, or Aspergillus sp., but was named Aspergillus brasiliensis, which raises questions. Since the strain isolated by the authors has significant differences both in the genomic sequence and in physiology associated with thermophilicity. Thus, A. brasiliensis CBS 101740 GCA_001889945.1 has a genome size of 35.8 Mb, and A. brasiliensis E_15.1 JBBEEP010000000 has a genome size of 36.9 Mb. The fact that the authors are dealing with a new species of Aspergillus is also indicated by the data in Figures 6, 7, and 10. In this regard, I would like to receive a detailed answer from the authors, based on what criteria they decided not to introduce a new species of Aspergillus, but to attribute the resulting isolate to Aspergillus brasiliensis. Otherwise, there are no comments on the work; it can be accepted for publication after minor corrections given below.

You are right. At first, we thought it was a new species since the numbers that you mention could be relevant. Also, the GC% content is a little bit different from the reference strain CBS 101740 GCA_001889945.1. The number of predicted proteins is also higher in E_15.1 than in the reference sequence. However interspecies GC values differences are expected to be between 1 and 2 %. In our case the value is only 0.74%. Regarding the genome size, Aspergilli show a big variability that goes between 20 to 40 Mb (approximately) and E_15.1 has only a 1.1 % difference with the reference strain.

Previous phylogenetic analysis with 13 A. brasiliensis strains, performed with three concatenated molecular markers (ITS, b-tubulin and calmodulin), placed E-15.1 in the A. brasiliensis clade (data not shown).

However, having the whole genome sequence allowed us to perform detailed state-of-the -art analyses that undoubtedly showed that this is a strain of A. brasiliensis. Probably the differences stated above are due to the stress conditions in which it was isolated (gene duplication, for example).

Analyses preformed using robust bioinformatic tools such as Orthofinder, Jolytree and UFCG to infer phylogenetic trees, resulted that E-15.1 groups with four complete genomes of A. brasiliensis strains deposited in the GenBank, forming a monophyletic group in all the analyzed phylogenies. Besides, speciation tests using bPTP and mPTP also showed that this strain is A. brasiliensis.

Additionally, genomic coherence measurements such as the global genomic relation index (OGRI) showed comparative and representative values for A. barslilensis in all cases. Values of ANI (Average Nucleotide Identity) and Mash genomic distances showed values above 95%, which is the threshold to determine if a strain belongs to that species. With E.15.1 we obtained values of 96.7% and 0.02 respectively, comparing strain E_15.1 and the genome of reference strain GCA001889945.1. Although this is mentioned in the manuscript, a table with this data has been included as supplementary material.

Still not fully convinced we submitted the E.1.5 genome to a k-mer analysis using the FOCUS software; this tool allows to detect alleles and mutations between species and in our case the obtained value was 90.7 % of identical k-mers between A. brasiliensis E_15.1 and A. brasiliensis IFM_66951, indicating that E_15.1 is indeed a A. brasiliensis species.

Detail comments

Lines 48-50

In the abstract you say that "The average temperature of the collection site was around 60 °C". In addition, you collected plant samples (Andropogon  sp. )from which the thermophilic fungus A. brasiliensis was extracted. Therefore, the host plants were exposed to the same thermophilic conditions, if not even more extreme. In this regard, it is not clear why you say that “only a limited number of fungal species are recognized as genuinely thermophilic”. What about the host organism? Why can't host plants also be called genuinely thermophilic? And the fact that they would not become thermophilic without an endophytic fungus also cannot exclude them from thermophilic, since endosymbiosis is one of the adaptation mechanisms.

You are right. It is very possible that the plant is also thermophilic. Unfortunately, we were unable to breed the plant in the laboratory, so we could not test its thermal preferences with and without the endophyte.

The reference to genuine thermophilic fungi is quoted according to the consulted literature. Thermophilic organisms are those that grow better at higher temperatures than related species. In this case, E-15.1 grew optimally at 40 ºC, while most Aspergilli are reported to grow optimally around 28 ºC. However, this strain provided tolerance to heat stress in a different plant from which it was isolated (S. lycopersicum) suggesting a broad host interaction with plants. Certainly, other plants species (thermophilic or not) should be tested to determine the range of interaction with this strain. Just as an example, Trichoderma species are not too specific regarding the plant species they can colonize.

Line 50.

Incorrect citation of a literary source in connection with The upper-temperature threshold for eukaryotes. The primary source is Tansey MR, Brock TD (1972) The upper temperature limit for eukaryotic organisms. Proc Natl Acad Sci U S A 69:2426–2428. And the indicated literary source itself cites the article Tansey MR, Brock TD (1972)

The reference refers to fungi, not all eukaryotes. As they are mentioned in a previous sentence, we will include the reference that you suggest. Thank you

Line 73

Incorrect citation number 7. (Ali, A.H.; Abdelrahman, M.; Radwan, U.; El-Zayat, S.; El-Sayed, M.A. Effect of Thermomyces Fungal Endophyte Isolated from 834 Extreme Hot Desert-Adapted Plant on Heat Stress Tolerance of Cucumber. Applied Soil Ecology 2018124, 155–162, 835 doi:10.1016/j.apsoil.2017.11.004.). This article does not describe Curvularia protuberate.

You are right, the title of the paper was wrong It has now been corrected:

Ali, A.H.; Abdelrahman, M.; Radwan, U.; El-Zayat, S.; El-Sayed, M.A. Desert plant-fungal endophytic association: the beneficial aspects to their hosts. Biological Forum-An International Journal 2018, 10, 138–145.

Line 75

Incorrect citation of a literary source, The primary source is - Redman RS, Sheehan KB, Stout RG, Rodriguez RJ, Henson JM (2002) Thermotolerance generated by plant/fungal symbiosis. Science 298:1581

This has been also corrected

[8] Redman, R.S.; Sheehan, K.B.; Stout, R.G.; Rodriguez, R.J.; Henson, J.M. Thermotolerance Generated by Plant/Fungal Symbiosis. Science 2002, 298, 1581, doi:10.1126/science.1072191.

Lines 111-113

Reword the sentence, in the current version you get that siderophores are a heat-stable enzyme, like phytases. And this is wrong, since siderophores are low-molecular compounds.

Thank you, you are rigth. Now ir reads:

…” these fungi can produce siderophores and heat-stable enzymes, such as phytases,

Move the sampling map (supplementary file 1) into the body of the article. Information related to the local location of an object is of interest to readers, and supplementary information is less frequently viewed than the main text of the article. Improve the design of the Figure. In its current form, its lower left part is empty.

The Figure has been included in the main text and modified to better use the space for it.

Figure 1

This figure is oversaturated with information, too many numbers, from A to O. This makes some information difficult to perceive. In particular, in Figure 1.O it is not clear which part of Figure corresponds to which temperature. Divide this Figure into two Figures, in the first one leave A-L, to the second Figure move M-O. This will make the material easier to understand.

Thank you for your suggestion, we have divided the figure in two separate figures (2 and 3), carefully revising the figure order and their mention in the text.

Line 415

In the current version of submission, the source you specified (Supplementary Table S2) is not attached. Only resource available: Dataset S1. NCBI non-redundant database annotation of E_15.1 predicted proteins

We have now made sure that all supplementary material is in the file. Thank you for your comments

Reviewer 3 Report

As described by the author, there have been very few research reports on thermophilic fungi so far. At present, the majority of thermophilic organisms are thermophilic bacteria or thermophilic archaea. Some characteristics of fungi are difficult for bacteria or archaea to replace, so the research results and new discoveries in this article have significant value. Aspergillus brasiliensis E15.1 not only has important applications in plant stress resistance research, but also serves as an important source of many thermophilic enzymes. If this fungus can achieve large-scale cultivation, it can even serve as an excellent chassis microorganism for high-temperature bioreactors, making it unique in the field of synthetic biology.

My main focus is on three points.

1. The similarity values between Aspergillus brasiliensis E15.1 and other strains in Figure 7 (JolyTree) are not clear, which reflects the novelty of this strain. Of course, the entire Figure 7 is not clear enough, it is recommended to replace it.

2. In Figure 13, using only one plant to demonstrate the effectiveness of plant growth promotion seems insufficient. It would be more convincing if there were more plant data available.

3. In Figure 14, the last four figures lack data on enzyme activity below 70 degrees Celsius. The relationship between normal enzyme activity and temperature requires a peak, indicating the optimal temperature for the enzyme. Similarly, the activity data of thermophilic enzymes at low temperatures is also important, as some thermophilic enzymes do not show a significant decrease in activity with decreasing temperature and have special uses. If these data are available, it is recommended to supplement them.

Author Response

Answers to reviewer 3 (in the word file, the colors of the text help to identify the answers)

As described by the author, there have been very few research reports on thermophilic fungi so far. At present, the majority of thermophilic organisms are thermophilic bacteria or thermophilic archaea. Some characteristics of fungi are difficult for bacteria or archaea to replace, so the research results and new discoveries in this article have significant value. Aspergillus brasiliensis E15.1 not only has important applications in plant stress resistance research, but also serves as an important source of many thermophilic enzymes. If this fungus can achieve large-scale cultivation, it can even serve as an excellent chassis microorganism for high-temperature bioreactors, making it unique in the field of synthetic biology.

Thank you for this comment, indeed it is an interesting and novel model.

My main focus is on three points.

  1. The similarity values between Aspergillus brasiliensis E15.1 and other strains in Figure 7 (JolyTree) are not clear, which reflects the novelty of this strain. Of course, the entire Figure 7 is not clear enough, it is recommended to replace it.

We agree that Figure 7 is not very clear, but this happens to any phylogenomic tree performed with a considerable number of strains to obtain solid results. This is why a box protrudes from the tree to show the detail (including the bTPT and mTPT analyses) of how E-15.1 belongs undoubtedly to A. brasiliensis. We also believe that the rest of the tree should remain because using the word processor tools any section can be amplified to see the relation of other strain or species in this phylogenomic analysis and this will be useful to the scientific community. The same happens with Figures 8 and 9.

  1. In Figure 13, using only one plant to demonstrate the effectiveness of plant growth promotion seems insufficient. It would be more convincing if there were more plant data available.

In Figure 13 the plant images are just photographs of representative individuals from each treatment. Each experiment was performed in triplicate with biological replicas. Below the plant photographs are the bar depiction of the means of each parameter with the statistical analysis depicted both with lines and asterisks and show that A. brasiliensis E_15.1 has a significative influence on these parameters: this is the last sentence on the figure caption: “This assay was performed in triplicate and statistical significance was assessed using a Tukey test (*p≤0.05, ** p˂0.01, statistical significance indicated by asterisks). Ns: non-significant. Scale bars=1cm”

  1. In Figure 14, the last four figures lack data on enzyme activity below 70 degrees Celsius. The relationship between normal enzyme activity and temperature requires a peak, indicating the optimal temperature for the enzyme. Similarly, the activity data of thermophilic enzymes at low temperatures is also important, as some thermophilic enzymes do not show a significant decrease in activity with decreasing temperature and have special uses. If these data are available, it is recommended to supplement them.

You are right. We are sorry because the Figure caption was incomplete. Now it reads:

“Figure 14. Effect of temperature on xylanase and cellulase activities of E_15.1 strain. The two upper panels show the optimal temperature for each kind of enzyme. The four lower panels show the thermostability of the cellulase and xylanase activities. The horizontal bars within each box represent the mean of the triplicates while the lines outside the boxes show the standard deviation. Different letters indicate significant differences determined by one-way ANOVA and Tukey's multiple mean comparison when p ≤ 0.05. Ns: non-significant. RA: remaining activity after incubation one hour at the depicted temperature”

So, in this figure two parameters are shown and we decided to put them in a single figure since the paper has already a large number of figures. The two upper panels measure the optimal temperature enzymatic activity by incubating the extracts for 15 minutes at a range of temperatures that go from 30 to 70 ºC. The four lower panels represent the same data in two manners: the middle panels show the actual residual activity in Units/ml, while the lower panels show the % of remaining activity after one hour incubation at high temperatures, taking as 100% the highest activity achieved. Since at 80 ºC we noticed a diminished activity for most cases, optimal temperature was not tested beyond 70 ºC
